# The Makorin *lep-2* and the lncRNA *lep-5* regulate *lin-28* to schedule sexual maturation of the *C. elegans* nervous system

Hannah Lawson[1], Edward Vuong[2], Renee M Miller[3], Karin Kiontke[4], David HA Fitch[4], Douglas S Portman[1,2,5,6]*

[1]Department of Biology, University of Rochester, Rochester, United States; [2]Department of Biomedical Genetics, University of Rochester, Rochester, United States; [3]Department of Brain and Cognitive Sciences, University of Rochester, Rochester, United States; [4]Center for Developmental Genetics, Department of Biology, New York University, New York, United States; [5]Department of Neuroscience, University of Rochester, Rochester, United States; [6]DelMonte Institute for Neuroscience, University of Rochester, Rochester, United States

**Abstract** Sexual maturation must occur on a controlled developmental schedule. In mammals, Makorin3 (*MKRN3*) and the miRNA regulators *LIN28A/B* are key regulators of this process, but how they act is unclear. In *C. elegans*, sexual maturation of the nervous system includes the functional remodeling of postmitotic neurons and the onset of adult-specific behaviors. Here, we find that the *lin-28–let-7* axis (the 'heterochronic pathway') determines the timing of these events. Upstream of *lin-28*, the Makorin *lep-2* and the lncRNA *lep-5* regulate maturation cell-autonomously, indicating that distributed clocks, not a central timer, coordinate sexual differentiation of the *C. elegans* nervous system. Overexpression of human *MKRN3* delays aspects of *C. elegans* sexual maturation, suggesting the conservation of Makorin function. These studies reveal roles for a Makorin and a lncRNA in timing of sexual differentiation; moreover, they demonstrate deep conservation of the *lin-28–let-7* system in controlling the functional maturation of the nervous system.
DOI: https://doi.org/10.7554/eLife.43660.001

*For correspondence:
douglas.portman@rochester.edu

**Competing interests:** The authors declare that no competing interests exist.

## Introduction

The timing of sexual maturation is subject to complex regulation that reflects competing demands on biological systems (*Bogin et al., 2011*; *Gluckman and Hanson, 2006*; *Parent et al., 2003*). Over evolutionary time, the pressure on a species to reproduce rapidly is countered by the need to allow sufficient time for robust juvenile development. Optimum balance of these pressures is encoded by internal, genetic mechanisms that guide the timing of the juvenile-to-adult transition (*Zhu et al., 2018*). At the level of an individual, internal and external stimuli—for example the presence of mates and food, as well as health and nutritional status—interact with these timers to determine the onset of sexual maturation (*Abreu and Kaiser, 2016*; *Avendaño et al., 2017*; *Livadas and Chrousos, 2016*; *Plant, 2015*).

In general, the nature of these genetic timers and their interaction with other signals is not well understood. Furthermore, while the hallmark of the juvenile-to-adult transition is the functional maturation of the germline and genitalia, reproductive maturation is typically accompanied by a much broader suite of changes in morphology, physiology, and behavior. These behavioral changes entail not only the activation of copulatory programs themselves, but also include modifications to

**eLife digest** Most animals develop from juveniles, which cannot reproduce, to sexually mature adults. The most obvious signs of this transition are changes in body shape and size. However, changes also take place in the brain that enable the animals to adapt their behavior to the demands of adulthood. For example, fully fed adult male roundworms will leave a food source to search for mates, whereas juvenile males will continue feeding.

The transition to sexual maturity needs to be carefully timed. Too early, and the animal risks compromising key stages of development. Too late, and the animal may be less competitive in the quest for reproductive success. Cues in the environment, such as the presence of food and mates, interact with timing mechanisms in the brain to trigger sexual maturity. But how these mechanisms work – in particular where and how an animal keeps track of its developmental stage – is not well understood.

In the roundworm species *Caenorhabditis elegans*, waves of gene activity, known collectively as the heterochronic pathway, determine patterns of cell growth as animals mature. Through further studies of these worms, Lawson et al. now show that these waves also control the time at which neural circuits mature. In addition, the waves of activity occur inside the nervous system itself, rather than in a tissue that sends signals to the nervous system. Moreover, they occur independently inside many different neurons. Each neuron thus has its own molecular clock for keeping track of development.

Several of the genes critical for developmental timekeeping in worms are also found in mammals, including two genes that help to control when puberty starts in humans. If one of these genes – called *MKRN3* – does not work correctly, it can lead to a condition that causes individuals to go through puberty several years earlier than normal. Studying the mechanisms identified in roundworms may help us to better understand this disorder. More generally, future work that builds on the results presented by Lawson et al. will help to reveal how environmental cues and gene activity interact to control when we become adults.

DOI: https://doi.org/10.7554/eLife.43660.002

decision-making circuits, allowing animals to incorporate reproductive drives when prioritizing behavioral programs. Whether genetic timers function at a single hub to coordinate behavioral transitions, or whether instead they may have distributed functions across different cell types, is unclear.

The timing of sexual maturation has been extensively studied in mammals, where it is coordinated by the temporally controlled production of gonadal steroids at puberty (*Abreu and Kaiser, 2016*; *Avendaño et al., 2017*; *Livadas and Chrousos, 2016*; *Plant, 2015*). This process is activated by the hypothalamic-pituitary-gonadal (HPG) axis, in which the pulsatile release of gonadotropin-releasing hormone (GnRH) by neurons in the hypothalamus activates gonadotropin production in the pituitary gland. This, in turn, triggers gonadal maturation and subsequent steroid hormone production. Upstream of HPG axis activation, a complex set of regulatory inputs converges on the release of the neuropeptide kisspeptin in the arcuate nucleus of the hypothalamus to stimulate GnRH production. Multiple studies suggest that kisspeptin release is influenced by environmental signals as well as internal timing cues (*Harter et al., 2018*). Recent studies in mice and humans have identified a number of genes important for this process, including the miRNA regulators *LIN28A/B*, their target *LET7*, and the Makorin *MKRN3*, whose loss causes Central Precocious Puberty in humans (*Abreu et al., 2013*; *Corre et al., 2016*; *Gajdos et al., 2010*; *Ong et al., 2009*; *Park et al., 2012*; *Yi et al., 2018*; *Zhu et al., 2010*; *Zhu et al., 2018*). However, where and how these genes act to regulate the onset of reproductive maturation remains unclear. Moreover, some aspects of this model are likely to be specific to mammals, highlighting the importance of studying these questions in diverse animals.

In the nematode *C. elegans*, juveniles pass through four larval stages before becoming sexually mature adults, offering an ideal opportunity to understand the mechanisms that regulate the juvenile-to-adult transition. Most studies addressing this question have focused on stage-specific events during larval development, particularly in the hypodermal seam cells, epidermal-like cells that lie along the sides of the body. In seam cells, stage-specific patterns of proliferation and differentiation

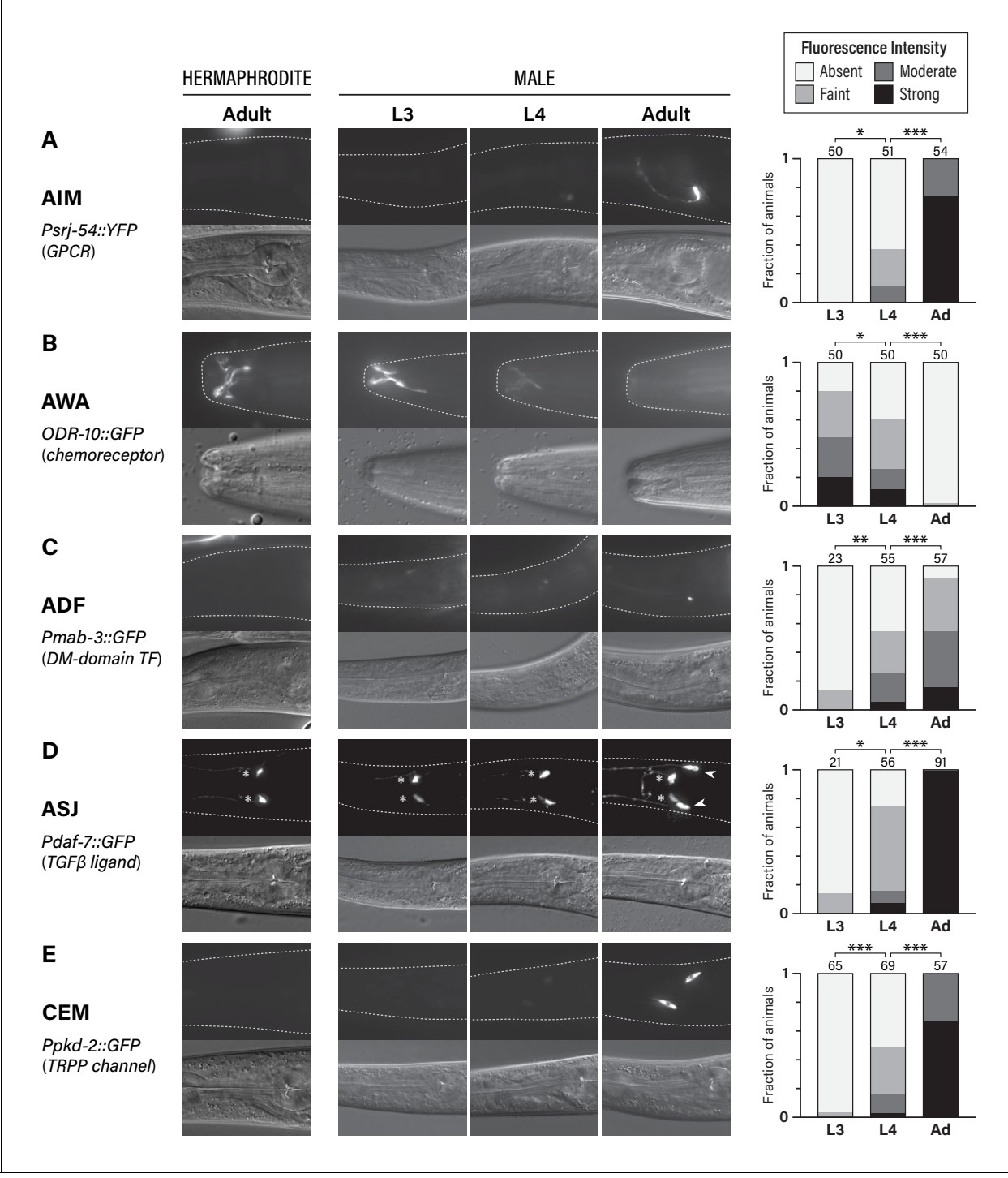

**Figure 1.** Five classes of neurons undergo male-specific functional maturation during the larval-to-adult transition. (**A–E**) Rows show expression of reporter genes for *srj-54*, *odr-10*, *mab-3*, *daf-7*, and *pkd-2*, respectively. At the left, the name of each marker, a brief description of its gene product, and the cell type whose maturation it marks are indicated. For all rows, representative epifluorescence (above; dashed line indicates outline of body) and DIC (below) images are shown for the indicated sex and stage. The graph at the right of each row shows expression levels in L3, L4, and adult males. Asterisks indicate the statistical significance of comparisons between L3 and L4 and between L4 and adult. Numbers above each bar indicate numbers of animals scored.

DOI: https://doi.org/10.7554/eLife.43660.003

The following source data is available for figure 1:

**Source data 1.** Source data for *Figure 1*.
DOI: https://doi.org/10.7554/eLife.43660.004

are controlled by a complex mechanism known as the 'heterochronic pathway' (*Rougvie and Moss, 2013*). In this mechanism, temporally controlled waves of miRNA expression (including the first two miRNAs discovered in any system, *lin-4* and *let-7*) act on a variety of targets to specify stage-specific events and regulate the transition from one stage to the next. The conserved miRNA regulator *lin-28* is also a key component of this mechanism. The heterochronic pathway also controls some events that occur during the larval-to-adult transition, including the cessation of molting, the fusion of the seam cells, and male-specific tail tip morphogenesis (TTM), a remodeling of four postmitotic posterior hypodermal cells (*Del Rio-Albrechtsen et al., 2006*; *Rougvie and Moss, 2013*). In some heterochronic mutants ('precocious' mutants), particular stage-specific events are skipped or occur too early; in others ('retarded' or 'delayed' mutants), events are delayed or absent.

Recent studies of TTM have identified two additional heterochronic genes, *lep-2*/Makorin and the lncRNA *lep-5*, that influence developmental timing by regulating LIN-28 stability (*Herrera et al., 2016*; *Kiontke et al., 2019*). Mutations in these genes cause a delayed phenotype, the retention of a larval-like, pointed ('Lep') tail tip in males. These mutants also have developmental defects in the body hypodermis, but they lack the seam cell lineage disruptions of other heterochronic mutants. Interestingly, *lep-2* mutant males have defects in mating behavior, suggesting a role for *lep-2* in the sexual maturation of the nervous system (*Herrera et al., 2016*).

While the maturation of the hypodermis has been well studied, the mechanisms that control the timing of adult-specific sexual behavior in *C. elegans* are poorly understood. In both sexes, males and hermaphrodites (somatic females that produce both eggs and sperm), sexual behavior requires larval neurogenesis that produces sex-specific circuits for copulation and egg-laying behavior (*Emmons, 2018*; *García and Portman, 2016*). The development of these circuits is independent of the gonad, relying instead on intrinsic lineage programs in which chromosomal sex acts via the sex-determination hierarchy to regulate neurogenesis and cell fate specification (*Barr et al., 2018*). Furthermore, multiple aspects of adult male behavior also require changes in synaptic connectivity and the functional modulation of pre-existing, post-mitotic neurons and circuits at the juvenile-to-adult transition (*Portman, 2017*). For example, the AWA olfactory neurons express the food-associated chemoreceptor *odr-10* in larvae of both sexes, but in males, *odr-10* expression is repressed at the larval-to-adult transition. This repression desensitizes males to food stimuli and promotes mate-searching behavior (*Ryan et al., 2014*). In other cases, adult-specific activation of gene expression in males, such as the expression of *daf-7* in the ASJ neurons (*Hilbert and Kim, 2017*) facilitates adult behavior. The extent to which sex-shared, postmitotic neurons are functionally modulated at the juvenile-to-adult transition remains unclear, and the mechanisms that control the timing of this modulation are unknown.

The striking involvement in mammalian puberty of orthologs of three *C. elegans* heterochronic genes (*lep-2*, *lin-28*, and *let-7*), together with the emerging appreciation that *C. elegans* sexual differentiation involves functional transitions in post-mitotic neurons, led us to explore the role of the heterochronic pathway in the timing of these changes. We find that the *lep-2/lep-5* branch of the heterochronic pathway plays a key role in timing the sexual differentiation of the *C. elegans* nervous system, demonstrating conservation of mechanism across species and raising the possibility that an unidentified *lep-5*-like lncRNA may play a key role in mammalian sexual maturation. Moreover, we find that this pathway acts cell-autonomously in multiple neurons, suggesting that genetic timing information is widely distributed across the nervous system. Multiple targets of *lin-41*, including *mab-3* and *lin-29a*, integrate the heterochronic and sex-determination pathways to regulate downstream effector genes and the activation of multiple adult behaviors. Interestingly, although the mammalian Makorin *MKRN3* functions in the opposite direction as its *C. elegans* ortholog *lep-2*, we find that *MRKN3* expression in the *C. elegans* nervous system is sufficient to delay sexual maturation, indicating that the pathway in which it interacts is functionally conserved across species. These findings identify the heterochronic pathway as an ancient regulator of sexual differentiation of the nervous system, provide specific mechanistic hypotheses for the functions of these genes in the mammalian brain, and suggest an important role for a yet-unidentified mammalian lncRNA in the timing of puberty.

## Results

### Multiple *C. elegans* neurons undergo male-specific functional maturation during the juvenile-to-adult transition

To visualize the sexual maturation of the *C. elegans* nervous system, we studied the developmental dynamics of the expression of five genes known to display sexually dimorphic expression in adults (*Figure 1*). Four of these reflect sexual dimorphism in shared (*i.e.*, non-sex-specific) neurons: *srj-54*, a GPCR of unknown function expressed male-specifically in the AIM interneurons (*Lee and Portman, 2007*; *Portman, 2007*); *odr-10*, a chemoreceptor whose male-specific downregulation in the AWA olfactory neurons is associated with decreased diacetyl and food detection (*Lee and Portman, 2007*; *Ryan et al., 2014*; *Sengupta et al., 1996*); *mab-3*, a *doublesex*-family transcription factor that is male-specifically expressed in the ADF chemosensory neurons, where it promotes the detection of hermaphrodite sex pheromone (*Fagan et al., 2018*; *Yi et al., 2000*); and *daf-7*, a TGFβ-superfamily ligand that is male-specifically expressed in the ASJ chemosensory neurons and promotes the food-leaving behavior of adult males (*Hilbert and Kim, 2017*; *Ren et al., 1996*). We also studied *pkd-2*, a marker for the male-specific CEM pheromone-detection neurons, which exist in an undifferentiated state until late larval development (*Barr and Sternberg, 1999*; *Narayan et al., 2016*; *Srinivasan et al., 2008*; *Sulston et al., 1980*; *Sulston et al., 1983*). Importantly, all of these neurons are generated during embryogenesis and, with the exception of the CEMs, are thought to be functional components of larval circuits.

For all of the markers we examined, we found that sex-specific expression patterns were absent in young larvae, emerging only late in larval development (*Figure 1*). In L3 males, we detected virtually no expression of *srj-54* in AIM, *mab-3* in ADF, or, consistent with a previous report (*Hilbert and Kim, 2017*), *daf-7* in ASJ. Moderate expression of these markers was observed in L4 males, but the full extent of sex differences in expression was not fully apparent until adulthood (*Figure 1A,C,D*). With regard to *odr-10* expression in AWA, we observed the opposite pattern, confirming our earlier findings (*Ryan et al., 2014*): reporter expression was readily detectable in L3 males, as it is in adult hermaphrodites, and diminished in males through L4 into adulthood (*Figure 1B*). Thus, at least with respect to these markers, the AWA, AIM, ADF, and ASJ neurons in larval males exist in a hermaphrodite-like 'ground state' before they acquire adult-specific characteristics during the juvenile-to-adult transition. The CEMs, which are absent in hermaphrodites, underwent maturation with similar timing. As expected (*Barr and Sternberg, 1999*), we detected no expression of *pkd-2* in L3 males; intermediate expression was observed in L4 animals and was further strengthened in day-one adults (*Figure 1E*). This timing is consistent with that previously reported for other CEM-specific genes (*Barr et al., 2001*; *Portman and Emmons, 2004*; *Wang et al., 2015*) as well as the cholinergic maturation of these neurons (*Pereira et al., 2015*). Together, these findings reinforce the idea that the larval-to-adult transition involves not only the incorporation of new male-specific neurons into the nervous system (*Barr et al., 2018*), but also the molecular maturation of multiple classes of pre-existing neurons.

### The *lin-28–let-7* regulatory module controls the timing of sexual maturation of the *C. elegans* nervous system

To ask whether the heterochronic pathway has a role in the timing of these molecular maturation events, we examined animals carrying mutations in *lin-28* and *let-7*, as these genes are at the core of the late heterochronic timer (*Rougvie and Moss, 2013*) (*Figure 2A,B*). In the hypodermal seam and the male tail tip hypodermis, *lin-28* mutants precociously execute late-larval patterns of proliferation, differentiation, and morphogenesis (*Ambros, 1989*; *Ambros and Horvitz, 1984*; *Herrera et al., 2016*; *Moss et al., 1997*); conversely, *let-7* mutants retain larval-specific fates and male tail morphology as adults (*Reinhart et al., 2000*; *Slack et al., 2000*; *Vadla et al., 2012*).

We found that gene expression dynamics in the nervous system were similarly altered in these mutants (*Figure 2C–G*). *lin-28* mutant males exhibited precocious activation of *srj-54* in AIM, with clear reporter expression in over half of L3 males and over 80% of L4 males. In contrast, the sexual maturation of AIM was delayed in *let-7* mutant males, with very little *srj-54* expression detectable in L4 and only moderate expression in one-day adults (*Figure 2C*). (Since *let-7* null mutants are inviable, these experiments used the hypomorphic allele *n2853*; null phenotypes would likely be stronger

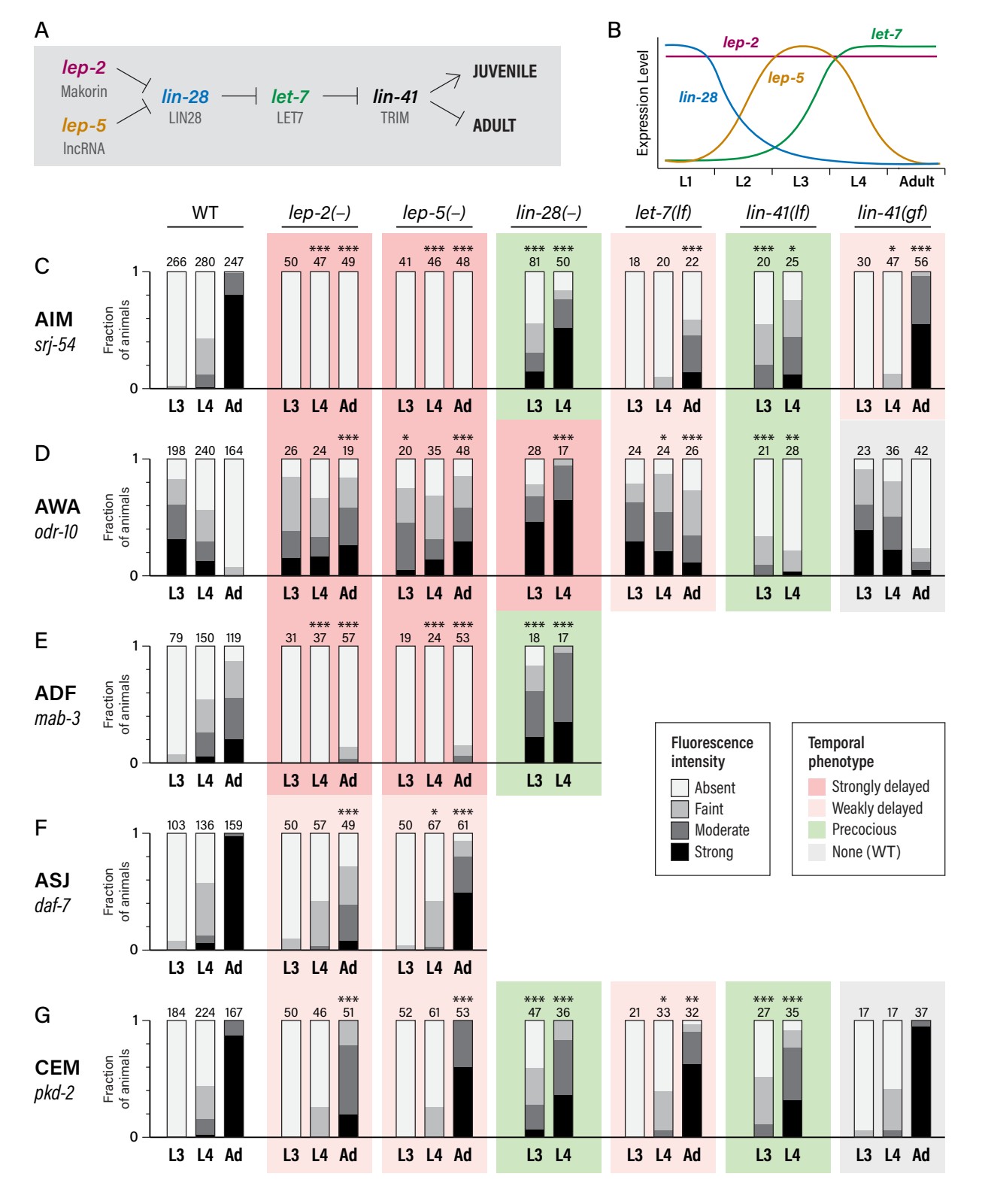

**Figure 2.** Heterochronic genes regulate the sexual maturation of multiple *C. elegans* neurons. (**A**) A subset of the heterochronic pathway, indicating genes and regulatory events important for the timing of the juvenile-to-adult transition in hypodermal tissues. (**B**) The temporal dynamics of *lep-2*, *lep-5*, *lin-28*, and *let-7* expression during larval development. (**C–G**) Expression of markers for the sexual maturation of five neuron classes in wild-type and heterochronic mutants. For each marker, expression is shown in the males of the indicated stage (L3, L4, and Adult) and genotype. '–" indicates that a

*Figure 2 continued on next page*

*Figure 2 continued*

null allele was used, while 'lf' and 'gf' indicate hypomorphic and hypermorphic alleles, respectively. For *lin-28(–)* and *lin-41(lf)*, only L3 and L4 males were scored. All strains were grown at 20℃ except for *let-7(lf)* and *lin-41(bx42)* and their paired wild-type controls, which were grown at 25℃. Asterisks indicate the statistical significance of comparisons to paired wild-type controls of the same stage. Paired wild-type control data was pooled to create the WT graph at the left of each row. Numbers above each bar indicate numbers of animals scored. Graphs are shaded to indicate temporal phenotypes as shown in the legend.

DOI: https://doi.org/10.7554/eLife.43660.005

The following source data and figure supplements are available for figure 2:

**Source data 1.** Source data for *Figure 2*.
DOI: https://doi.org/10.7554/eLife.43660.008

**Figure supplement 1.** Structural features of the *lep-5* lncRNA required for its function in the nervous system.
DOI: https://doi.org/10.7554/eLife.43660.006

**Figure supplement 1—source data 1.** Source data for *Figure 2—figure supplement 1*.
DOI: https://doi.org/10.7554/eLife.43660.007

(*Reinhart et al., 2000*).) We observed similar effects of *lin-28* and *let-7* mutations on *pkd-2* reporter expression in the CEMs (*Figure 2G*): loss of *lin-28* resulted in precocious expression in L3 males, while loss of *let-7* slightly reduced *pkd-2* expression in young adults. *let-7* mutant adults also retained larval-like expression of *odr-10* in AWA. Unexpectedly, however, *odr-10* downregulation did not occur precociously in *lin-28* mutant males; instead, levels increased in L4 (*Figure 2D*). Because *odr-10* downregulation does depend on genes downstream of *lin-28* (see below), some adult-specific characteristic(s) that are independent of *lin-28* are likely to be required for *odr-10* downregulation (*e.g.,* signals from male-specific components of the adult nervous system). Finally, we also observed precocious ADF expression of *mab-3* in *lin-28* mutant males (*Figure 2E*), with strong expression initiating in L3. Together, these results demonstrate that the *lin-28*–*let-7* regulatory axis governs the timing of key events in the sexual maturation of the *C. elegans* nervous system.

In the hypodermis, a key target of *let-7* is *lin-41*, which encodes an RBCC protein of the NHL family (*Slack et al., 2000*). We found that loss of *lin-41* function resulted in precocious activation of *srj-54* in AIM and *pkd-2* in CEM, and precocious loss of *odr-10* in AWA (*Figure 2C,D,G*). Reciprocally, a *lin-41(gf)* allele (*Del Rio-Albrechtsen et al., 2006*) caused somewhat delayed maturation of the expression of *srj-54*, though it had no statistically significant effects on *odr-10* or *pkd-2* expression (*Figure 2C,D,G*). This discrepancy could reflect cell-type-specific thresholds for the degree of *lin-41* function necessary to elicit a gain-of-function phenotype. Together, these results indicate that *lin-41* is an important effector of *let-7* function in the nervous system, as it is in the hypodermis.

## The makorin *lep-2* and the lncRNA *lep-5* act upstream of *lin-28* to promote sexual differentiation

In the timing of male tail tip morphogenesis, the Makorin *lep-2* and the lncRNA *lep-5* act upstream of *lin-28* (*Herrera et al., 2016*; *Kiontke et al., 2019*). Because loss-of-function mutations in the *lep-2* ortholog *MKRN3* cause Central Precocious Puberty in humans (*Abreu et al., 2013*) and because of the sexual behavior defects in *C. elegans lep-2* mutant males (*Herrera et al., 2016*), we considered the possibility that the *lep-2/lep-5* branch of the heterochronic pathway might function in sexual differentiation of the *C. elegans* nervous system. Consistent with this idea, we found that these animals exhibited striking defects in neuronal maturation: for *srj-54*, *odr-10*, and *mab-3*, day one adult *lep-5* males retained the expression pattern typically seen in wild-type L3 males (*Figure 2C,D,E*). We observed weaker phenotypes for *daf-7* and *pkd-2*: in both cases, expression increased during larval development, but failed to reach the level typically seen in young adults (*Figure 2F,G*). Thus, *lep-2* is required for the sexual differentiation of AIM, AWA, and ADF, and also contributes to this process in ASJ and CEM. Intriguingly, the absence of sexually mature characteristics in young *lep-2* adult males stands in contrast to the Central Precocious Puberty that results from loss of human *MKRN3* function, a distinction to which we return below.

As is the case for tail tip morphogenesis (*Kiontke et al., 2019*), the *lep-5* mutant phenotype almost completely phenocopied that of *lep-2*. Mutant males failed to adopt adult-specific gene expression patterns in AIM, AWA, and ADF, but had only modest defects in marker expression in ASJ and CEMs (*Figure 2C–G*). *lep-5* encodes a lncRNA that is predicted to adopt a compact

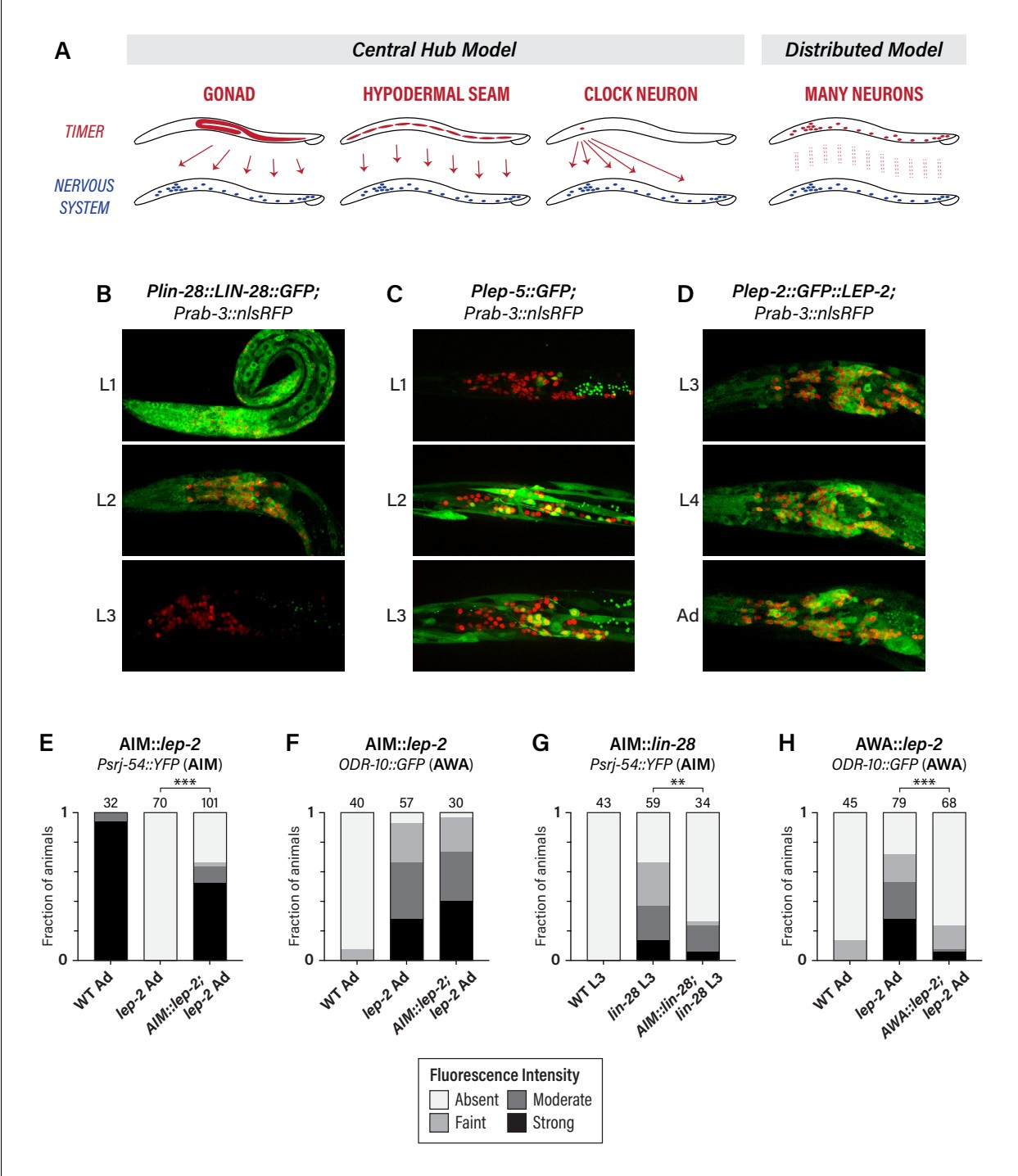

**Figure 3.** The heterochronic pathway is a cell-autonomous, distributed timer of sexual maturation. (A) Models for non-autonomous and autonomous functions of the heterochronic pathway, as described in the text. (B) Expression of *LIN-28::GFP* and the pan-neural nuclear marker *Prab-3::nRFP*. Strong GFP expression is detected throughout the nervous system of L1 animals; this decreases during L2 and is virtually undetectable in L3. (C) *Plep-5::GFP* expression in L1, L2, and L3 animals. GFP is essentially undetectable in L1 but is seen in many head neurons in L2 and L3. (D) *Plep-2::GFP::LEP-2* expression in L3, L4, and Adult males. Broad GFP expression is seen in the nervous system at all stages examined. (E–H) Cell-type specific rescue. Numbers above each bar indicate numbers of animals scored. Asterisks indicate the significance of the comparison between transgenic animals and non-transgenic control siblings. (E) Expression of *Psrj-54::YFP* in wild-type, *lep-2* (non-transgenic siblings), and *lep-2; AIM::lep-2* males of the indicated stage. (F) Expression of *ODR-10::GFP* in wild-type, *lep-2* (non-transgenic siblings), and *lep-2; AIM::lep-2* males of the indicated stage. (G) Expression of *Psrj-54::YFP* in wild-type, *lin-28* (non-transgenic siblings), and *lin-28; AIM::lin-28* males of the indicated stage. (H) Expression of *ODR-10::GFP* in wild-type, *lep-2* (non-transgenic siblings), and *lep-2; AWA::lep-2* adult males.

*Figure 3 continued on next page*

eLIFE Research article

Developmental Biology | Neuroscience

*Figure 3 continued*

DOI: https://doi.org/10.7554/eLife.43660.009

The following source data and figure supplement are available for figure 3:

**Source data 1.** Source data for *Figure 3*.
DOI: https://doi.org/10.7554/eLife.43660.011
**Figure supplement 1.** *lin-4* is necessary for proper *lep-5::GFP* expression in the nervous system.
DOI: https://doi.org/10.7554/eLife.43660.010

secondary structure with prominent stem-loop structures near its 5′ and 3′ ends, as well as a two-part central double-stranded 'zipper' region (*Figure 2—figure supplement 1A*) (*Kiontke et al., 2019*). *lep-5* has been proposed to function as an RNA scaffold, forming a tripartite complex with LEP-2 and LIN-28 to promote LIN-28 degradation (*Kiontke et al., 2019*). We found that a Crispr-mediated deletion of the predicted 3′ hairpin, *lep-5(fs20)*, phenocopied the *lep-5* null allele, showing a nearly complete lack of *srj-54* expression in AIM in adult males (*Figure 2—figure supplement 1B*). *lep-5(fs8)*, a point mutation that disrupts the 5′ hairpin, as well as *lep-5(fs21)*, a Crispr-engineered allele bearing a cluster of five point mutations predicted to disrupt the central zipper, caused a complete loss of *srj-54* expression in adults (*Figure 2—figure supplement 1B, C*). Furthermore, *lep-5 (fs21fs25)*, a double mutant that carries a set of compensatory changes predicted to restore the central zipper, completely restored adult expression of *srj-54* (*Figure 2—figure supplement 1B, C*). The phenotypes of these mutants are consistent with those seen in male tail tip morphogenesis (*Kiontke et al., 2019*), indicating that *lep-5* likely acts through a similar mechanism in both tail tip morphogenesis and neuronal maturation.

## The *lep-2–lin-28–let-7* axis acts cell-autonomously to time the onset of sexual maturation in the nervous system

We considered several scenarios by which heterochronic genes could control the timing of neuronal maturation (*Figure 3A*). In one, heterochronic genes might act in the gonad to control the secretion of hormonal signals. However, this seems unlikely, as the development of the *C. elegans* gonad is largely independent of the heterochronic pathway (*Rougvie and Moss, 2013*) and, with the exception of vulva and hindgut development, (*Kimble, 1981*; *Sulston et al., 1980*), the gonad does not play a significant role in the sexual differentiation of the *C. elegans* soma (*Barr et al., 2018*). Alternatively, the heterochronic pathway might act as a central somatic timer (*e.g.*, in seam cells or a master 'clock' neuron) from which it broadcasts temporal information to the nervous system. Finally, individual neurons might possess their own timers, such that temporal control is distributed throughout the nervous system.

To address this, we first confirmed earlier reports suggesting that *lin-28, lep-2,* and *lep-5* are broadly expressed in the nervous system (*Herrera et al., 2016*; *Kiontke et al., 2019*; *Moss et al., 1997*). Using a translational *LIN-28::GFP* reporter, we observed widespread expression in L1 and L2 animals, including extensive expression in the nervous system (*Figure 3B*). LIN-28::GFP abundance decreased during L2 in the nervous system and elsewhere, and by L3, expression was virtually undetectable, consistent with previous reports (*Moss et al., 1997*). We observed an inverse pattern of P*lep-5::GFP* expression in the nervous system, consistent with its previously described dynamics in the hypodermis (*Kiontke et al., 2019*). P*lep-5::GFP* was detectable in very few neurons in L1 larvae, but broad neuronal expression was apparent by L2 and L3 (*Figure 3C*). Interestingly, *lep-5* reporter expression appears to be restricted to a subset of the nervous system (*Figure 3C*), an important issue for future work. With a P*lep-2::lep-2*::GFP translational reporter (*Herrera et al., 2016*), we observed expression in many head neurons in early larvae (not shown) and in L3, L4, and adult males (*Figure 3C*). Because *lep-2* expression levels remain essentially constant during larval development (*Herrera et al., 2016*), *lep-2* likely does not itself provide timing information but rather seems to provide a permissive cue for the transition into adulthood. Instead, *lep-5* likely provides an instructive signal important for determining the onset of the sexual differentiation of the nervous system, as it does in the hypodermis (*Kiontke et al., 2019*).

To ask whether the heterochronic pathway functions in the nervous system itself, we restored expression of wild-type *lep-2* or *lin-28* in the nervous system of *lep-2* or *lin-28* mutants. Expression

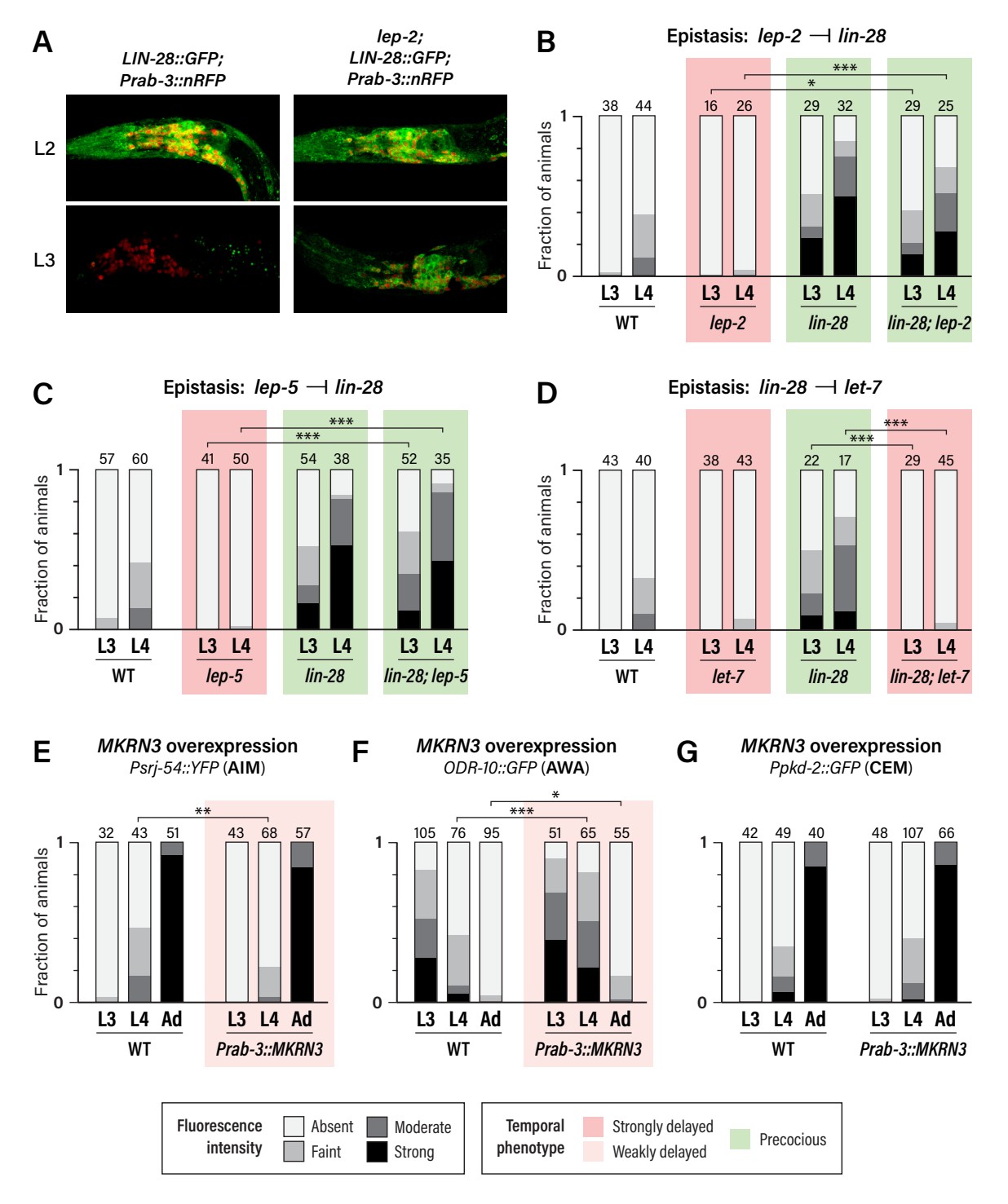

**Figure 4.** Regulatory relationships and conservation of heterochronic function. (**A**) *LIN-28::GFP* expression persists into L3 in *lep-2* mutants (right), unlike wild-type (left). The images on the left are the same as those in *Figure 3C*, but the GFP signal intensity is doubled to highlight the difference in expression in L3. Images for wild-type and mutant were taken with identical settings and images were processed identically. (**B–D**) *Psrj-54::YFP* expression in the indicated single and double mutant adult males. Numbers above each bar indicate numbers of animals scored. (**E–G**) *MKRN3* overexpression in the nervous system. Non-transgenic siblings were used as WT controls. Numbers above each bar indicate numbers of animals scored. Asterisks indicate significance of the comparison of transgenic animals to control animals of the same stage. (**E**) *Psrj-54::YFP* expression in animals of the indicated genotype and stage. (**F**) *ODR-10::GFP* expression in animals of the indicated genotype and stage. (**G**) *Ppkd-2::GFP* expression in animals of the indicated genotype and stage.

*Figure 4 continued on next page*

*Figure 4 continued*

DOI: https://doi.org/10.7554/eLife.43660.012

The following source data and figure supplements are available for figure 4:

**Source data 1.** Source data for *Figure 4*.

DOI: https://doi.org/10.7554/eLife.43660.015

**Figure supplement 1.** Genetic control of male sexual differentiation and behavior.

DOI: https://doi.org/10.7554/eLife.43660.013

**Figure supplement 1—source data 1.** Source data for *Figure 4—figure supplement 1*.

DOI: https://doi.org/10.7554/eLife.43660.014

of *lep-2* under the control of the AIM-specific promoter *Peat-4prom11* (*Serrano-Saiz et al., 2017*) restored *srj-54* expression to *lep-2* mutant adults, but had no effect on *odr-10* expression in AWA (*Figure 3E,F*), indicating that *lep-2* acts cell-autonomously to control timing in AIM. Furthermore, *Peat-4prom11::lin-28* rescued the precocious maturation of AIM in *lin-28* mutant larvae (*Figure 3G*). Thus, timing information intrinsic to AIM itself is critical for the timely control of AIM's maturation. This property is not limited to AIM, as AWA-specific expression of *lep-2* (*Pgpa-4Δ6::lep-2(+)*) rescued the lack of *odr-10* downregulation in AWA in *lep-2* mutants (*Figure 3H*). Together, these findings indicate that the heterochronic pathway does not control an organism-wide temporal signal that is broadcast from a central timer; rather, timing information is distributed, such that individual neurons maintain their own internal clocks.

### *lep-2* and *lep-5* function upstream of *lin-28* and its miRNA target *let-7* in the sexual differentiation of the nervous system

Multiple studies have shown that *lin-28* negatively regulates the biogenesis and activity of multiple miRNA targets, including *let-7*, in *C. elegans*, *Drosophila*, and mammals (*Tsialikas and Romer-Seibert, 2015*). The decay of *lin-28* activity is a key regulatory point in the progression into adulthood, allowing the activation of *let-7* and the subsequent downregulation of its targets. In the hypodermal seam, *lin-28* decay is controlled at both the mRNA and protein levels (*Huang et al., 2011*; *Morita and Han, 2006*; *Seggerson et al., 2002*; *Weaver et al., 2017*; *Weaver et al., 2014*). In the male tail tip, *lep-2* and *lep-5* function together to promote the degradation of LIN-28 protein during L3 (*Herrera et al., 2016*; *Kiontke et al., 2019*). We carried out a series of genetic and molecular epistasis experiments to ask which aspects of this regulatory logic extend to the nervous system.

Unlike wild-type males, in which neuronal LIN-28::GFP decays during L2 and is virtually absent by L3, *lep-2* mutant males exhibited strong neuronal LIN-28::GFP that persisted into L3 and L4 (*Figure 4A* and data not shown). Thus, consistent with its role in the tail tip and other hypodermal cells (*Herrera et al., 2016*), *lep-2* promotes the timely decay of LIN-28 in the nervous system. We also examined expression of the AIM maturation marker *srj-54* in *lin-28; lep-2* double mutants. Like *lin-28* single mutants, these animals exhibited precocious AIM maturation (*Figure 4B*), confirming that *lep-2* acts genetically upstream of *lin-28*. We observed a similar phenotype in *lin-28; lep-5* double mutants (*Figure 4C*), indicating that *lep-5* also acts upstream of *lin-28* with respect to nervous system maturation. Finally, we confirmed that *lin-28* acts upstream of *let-7* in the nervous system by examining *srj-54* expression in *lin-28; let-7* mutants. As expected, AIM maturation was delayed compared to wild-type (*Figure 4D*). Thus, *lep-2* and *lep-5* regulate the *lin-28–let-7* regulatory module to control the onset of sexual differentiation of the *C. elegans* nervous system.

Together with previous work (*Kiontke et al., 2019*), our results support the idea that the onset of *lep-5* expression is a critical step in determining the timing of LIN-28 decay and the subsequent progression into adulthood. To investigate mechanisms controlling the timing of *lep-5* activation, we considered a role for genes involved in the timing of early larval development. The miRNA *lin-4* and its key target, the transcription factor *lin-14*, regulate stage-specific development in L1 and L2 (*Rougvie and Moss, 2013*). We found that neuronal expression of P*lep-5::GFP* was significantly disrupted in *lin-4* mutants: very little reporter expression was detectable in the nervous system of larvae of any stage (*Figure 3—figure supplement 1*). This suggests that *lin-4* is necessary to trigger the onset of *lep-5* expression in the nervous system, consistent with its role in promoting the L1 to L2 transition (*Rougvie and Moss, 2013*).

## Human MKRN3 can inhibit sexual differentiation of the *C. elegans* nervous system

Because the *lep-2* and *lin-28* orthologs *MKRN3* and *LIN28A/B* are both implicated in the timing of human puberty, our results and those of *Herrera et al. (2016)* raise the possibility that *MKRN3* might act upstream of *LIN28A/B* in the mammalian nervous system. However, while loss of the *C. elegans* Makorin *lep-2* causes a delay in sexual differentiation, loss-of-function mutations in *MKRN3* are associated with the opposite phenotype, Central Precocious Puberty (*Abreu et al., 2013*). Several models can explain this apparent discrepancy. For example, it could be the case that both MKRN3 and LEP-2 inhibit LIN28A/B and LIN-28, but LIN28A/B might have multiple functions in mammalian puberty, acting both to promote and inhibit its onset (*Corre et al., 2016*). Alternatively, MKRN3 might act to stabilize LIN28A/B, possibly by inhibiting another mammalian Makorin that might function more like *C. elegans lep-2*. Finally, MKRN3 might regulate other substrates that function downstream of, or in parallel with, mammalian LIN28A/B in regulating puberty.

The first model predicts that *MKRN3* might be able to functionally substitute for *lep-2*. However, we found that pan-neural expression of *MKRN3* in *lep-2* mutants had no apparent effect on *srj-54* expression (*Figure 4—figure supplement 1A*). Though we cannot rule out the possibility that technical issues hindered rescue, this result suggests that LEP-2 and MKRN3 do not have equivalent biochemical functions.

Because wild-type *MKRN3* inhibits sexual differentiation in humans, we next asked whether it might have the same effect in *C. elegans*. Remarkably, expression of *MKRN3* from a pan-neuronal promoter in a wild-type background caused a modest but statistically significant delay in the activation of *srj-54* expression in AIM and the downregulation of *odr-10* expression in AWA (*Figure 4E,F*). However, we observed no effect of *MKRN3* overexpression on *pkd-2* expression (*Figure 4G*). The observation that *MKRN3* retains its ability to inhibit some aspects of sexual maturation when expressed in the *C. elegans* nervous system raises the intriguing possibility that *MKRN3* exerts an inhibitory effect on the conserved *lin-28–let-7*-dependent heterochronic timer and, moreover, that there is deep functional conservation of the mechanism in which it participates. The lack of an apparent effect of *MKRN3* overexpression on CEM maturation is consistent with our finding that these neurons rely less on *lep-2* and *lep-5* function (*Figure 2G*), indicating that their maturation occurs through mechanisms that are at least partially distinct from those that operate in AWA and AIM.

## Multiple effectors connect *lin-41* to the sexual maturation of the male nervous system

Our survey of heterochronic mutants (*Figure 2*) indicated that the *lin-28–let-7* pathway controls sexual maturity via the RNA-binding protein LIN-41, a direct target of *let-7* (*Slack et al., 2000*). LIN-41 regulates gene expression by binding to the 5′ or 3′ UTR of multiple target genes, including *lin-29a*, *mab-3*, *dmd-3*, and *mab-10* (*Aeschimann et al., 2017*). In the case of ADF, whose maturation is marked by the activation of *mab-3*, this provides a continuous molecular pathway connecting the *lep-2–lep-5–lin-28* timer to the sexual differentiation of ADF. With regard to the other neurons we studied, *mab-3* and *lin-29a* appear to have overlapping but distinct roles in sexual maturation. Studies carried out in parallel with this work found that *lin-29a* regulates expression of *srj-54* in AIM and *daf-7* in ASJ (*Pereira et al., 2019*). We found that the full extent of *srj-54* expression in AIM also depended on *mab-3* (*Figure 4—figure supplement 1B*). Thus, both *lin-29a* and *mab-3* promote AIM maturation, though the role of *mab-3* may be secondary and/or indirect. Furthermore, we found that the adult-specific repression of *odr-10* in AWA was lost in *mab-3* mutants but was unaffected by loss of *lin-29a* (*Figure 4—figure supplement 1C,D*). Thus, with respect to the molecular maturation events we have investigated here, both *mab-3* and *lin-29a* link the heterochronic pathway to downstream effectors of sexual differentiation. Because both *mab-3* and *lin-29a* are also targets of the master sexual regulator *tra-1* (*Fagan et al., 2018*; *Pereira et al., 2019*; *Yi et al., 2000*), these genes integrate sexual and temporal information to specify male-specific aspects of the sexual maturation of the *C. elegans* nervous system.

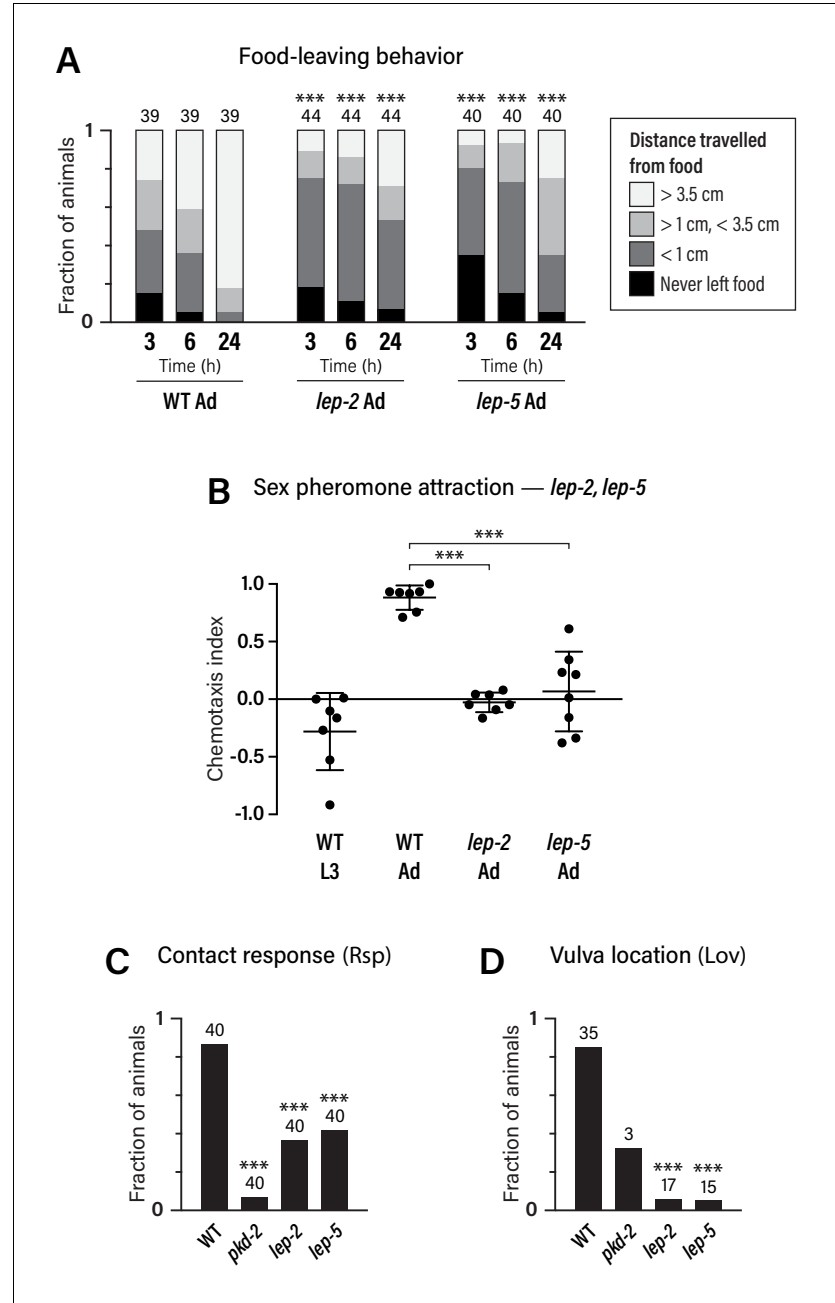

**Figure 5.** Adult-specific male behavior requires the function of the heterochronic pathway. (**A**) Food-leaving behavior. The maximum distance that a single animal had travelled from a small food spot was determined at 3, 6, and 24 hr after the beginning of the assay. Numbers indicate number of animals assayed. Asterisks indicate significance of comparisons to wild-type adults. (**B**) Attraction to a blend of ascaroside pheromones (ascr#2/#3/#8) using a quadrant-style assay. Each point represents the chemotaxis index calculated from one assay (ten animals). (**C, D**) Contact-response (Rsp) and Vulva-location (Lov) behavior. Numbers indicate number of animals tested. *pkd-2* mutant males, known to be defective in both behaviors (***Barr et al., 2001***), are used as controls.

DOI: https://doi.org/10.7554/eLife.43660.016

The following source data is available for figure 5:

**Source data 1.** Source data for *Figure 5*.

DOI: https://doi.org/10.7554/eLife.43660.017

## The heterochronic neuronal timer is necessary for the onset of sex differences in adult *C. elegans* behavior

With the exception of *srj-54*, whose function is unknown, each of the molecular markers used in this work is associated with adult-specific male behavior. Indeed, previous research has found that *lep-2* mutants have defects in male mating and food-leaving behaviors (*Herrera et al., 2016*), although the underlying mechanisms were unclear. We therefore surveyed these and other aspects of adult male behavior in both *lep-2* and *lep-5* mutants. While many heterochronic mutants have drastic, pleiotropic effects that interfere with behavioral assays, *lep-2* and *lep-5* males are, with the exception of a moderate defect in tail tip retraction, morphologically wild type. The retention of a pointed tail tip into adulthood is in itself thought not to significantly compromise male sexual behavior (*Del Rio-Albrechtsen et al., 2006*).

Food-leaving behavior, the propensity of well-fed males to leave a food source in search of mates, is manifested only in adulthood (*Lipton et al., 2004*). Multiple mechanisms promote male food-leaving, including signals from the male-specific ray sensory neurons, the neuropeptide receptor PDFR-1, the TGFβ-superfamily ligand DAF-7, and downregulation of the food-associated chemoreceptor *odr-10* (*Barrios et al., 2012*; *Barrios et al., 2008*; *Hilbert and Kim, 2017*; *Lipton et al.,*

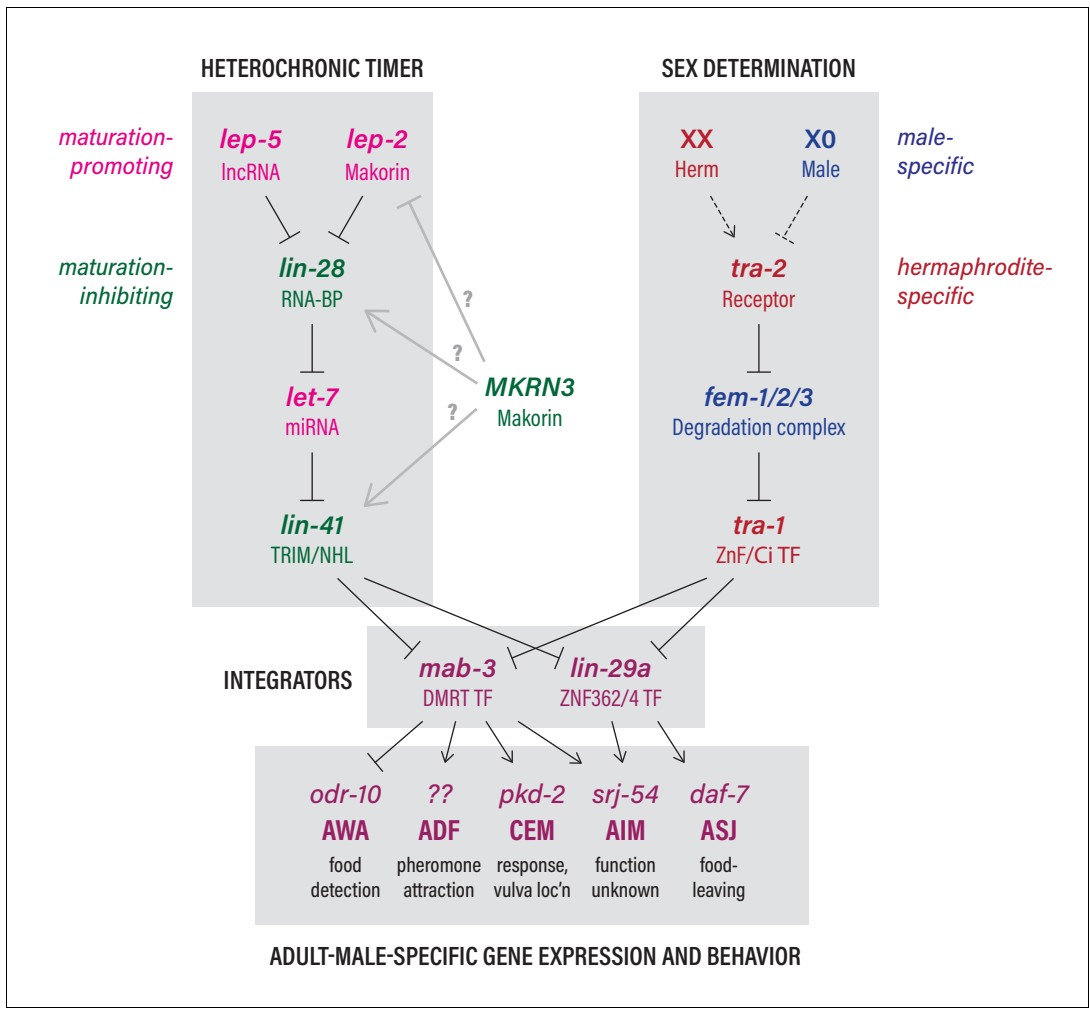

**Figure 6.** The heterochronic and sex-determination pathways intersect to control adult-specific male gene expression and behavior. Genes are color-coded to indicate function as indicated. Grey arrows from *MKRN3* represent hypothetical interactions that could explain the ability of *MKRN3* to inhibit *C. elegans* sexual differentiation. See text for details.
DOI: https://doi.org/10.7554/eLife.43660.018

*2004*; *Ryan et al., 2014*). We found that *lep-2* and *lep-5* mutant males exhibited markedly decreased food-leaving behavior as adults (*Figure 5A*), indicating that both of these genes are important for the acquisition of male sexual drive.

We also examined attraction to ascaroside sex pheromones, a male-specific behavior that depends partially on *mab-3* function in ADF (*Fagan et al., 2018*; *Srinivasan et al., 2008*). We found that ascaroside attraction was absent in male larvae: L3 males were slightly repelled by an ascr#2/ #3/#8 blend, reminiscent of the behavior of adult hermaphrodites (*Fagan et al., 2018*) (*Figure 5B*). The transition to the adult behavioral state requires *lep-2* and *lep-5*, as pheromone attraction was abolished in these mutants (*Figure 5B*). In contrast, *lin-29a* mutant males exhibited no apparent defects in ascaroside attraction (*Figure 3—figure supplement 1D*), suggesting that *mab-3* is the primary effector of the sexual maturation of ADF.

Finally, we examined two sub-steps of male mating, response behavior ('Rsp', the typical backward-locomotion response of males to physical contact with hermaphrodites) and the subsequent vulva-location behavior ('Lov', the ability of the male to locate the hermaphrodite vulva) (*Liu and Sternberg, 1995*). While these behaviors depend primarily on male-specific sensorimotor circuits in the tail, sex-specific characteristics of shared neurons are also likely to be important for their execution (*Barr and Sternberg, 1999*; *Koo et al., 2011*; *Liu and Sternberg, 1995*; *Liu et al., 2011*; *Sherlekar et al., 2013*) (R.M.M. and D.S.P., unpublished results). We found that response behavior was reduced in both *lep-2* and *lep-5* mutant males, while vulva-location behavior was almost completely absent (*Figure 5C,D*). Together, these findings demonstrate that the Makorin *lep-2* and the lncRNA *lep-5*, and by extension, the heterochronic pathway, are important for implementing adult-specific sexual behavior in *C. elegans* males.

## Discussion

Although it has long been appreciated that animal behavior undergoes functional transitions commensurate with sexual maturation (*Spear, 2000*), the mechanisms that determine the timing of these changes are poorly understood. Indeed, in humans, adolescence is a period of heightened susceptibility to a variety of neuropsychiatric disorders (*Jones, 2013*; *Walker et al., 2017*), highlighting the importance of understanding the regulatory mechanisms that control these transitions. Here, we unite disparate findings from studies of mammalian puberty and *C. elegans* developmental biology to establish that a conserved developmental timer functions cell-autonomously to coordinate the sexual maturation of the *C. elegans* nervous system (*Figure 6*). Central to this mechanism is the temporal control provided by the lncRNA *lep-5*, which, together with the Makorin *lep-2*, promotes the timely decay of the conserved miRNA inhibitor LIN-28 (*Kiontke et al., 2019*). This decay permits the biosynthesis of mature *let-7* miRNA and, through the pathway shown in *Figure 6*, promotes the juvenile-to-adult transition.

The maturation events we have studied here entail not the incorporation of new neurons and circuits into the nervous system, but rather gene expression and functional changes in pre-existing, postmitotic neurons. As such, these events parallel similar changes in the mammalian hypothalamus, where both gene expression and physiological changes determine the timing of HPG axis maturation and the onset of puberty (*Abreu and Kaiser, 2016*; *Avendaño et al., 2017*; *Livadas and Chrousos, 2016*; *Plant, 2015*). Because *MKRN3*, *LIN28A/B*, and *LET7* are expressed in the hypothalamus and elsewhere in the mammalian brain, our results indicate that the neuronal function of the heterochronic pathway is an ancient regulator of the juvenile-to-adult transition. In this view, the activational effects of gonadal steroids at puberty (*McCarthy et al., 2012*) are likely to be a recent evolutionary adaptation that reinforces sexual differentiation of the mammalian nervous system via an endocrine feedback loop. Moreover, it suggests that puberty-independent events outside the hypothalamus that contribute to behavioral maturation during mammalian adolescence, such as the transition from play to aggression in Siberian hamsters (*Paul et al., 2018*), could also be timed by the heterochronic pathway.

In addition to identifying neuronal functions for *lep-2*, *lin-28, and let-7*, we also show that the recently identified lncRNA *lep-5* functions in this pathway. *lep-5* was identified in forward genetic screens for male tail morphogenesis defects; in *lep-5* mutants, LIN-28 protein persists longer than in wild-type, delaying the onset of expression of *dmd-3*, the master regulator of tail tip morphogenesis (*Kiontke et al., 2019*). Because *lep-5* is expressed in a temporal wave whose rise corresponds to the

time at which *lin-28* expression decays, *lep-5* is thought to provide an instructive temporal cue for the destabilization of LIN-28 and, in turn, allow production of the *let-7* miRNA and the progression through late-larval development. Because *lep-2* mutants have a phenotype essentially identical to that of *lep-5*, *Kiontke et al., 2019* have proposed that *lep-5* serves as an RNA scaffold that recruits both LEP-2 and LIN-28; such a model is supported by in vivo crosslinking experiments demonstrating that *lep-5* binds to both LEP-2 and LIN-28. As a member of the Makorin family, LEP-2 may possess both RNA-binding and putative E3 ubiquitin ligase activities (*Herrera et al., 2016*), suggesting that a tripartite LEP-2–*lep-5*–LIN-28 complex could destabilize LIN-28 through ubiquitination by LEP-2 (*Kiontke et al., 2019*). Sequence homologs of *lep-5* have not been found outside of nematodes; however, lncRNAs are known to evolve rapidly, usually making it impossible to identify orthologs based on primary sequence alone (*Diederichs, 2014*). It is tempting to speculate that structural and/ or functional orthologs of *lep-5* are present in mammals and could have an important role in the regulation of LIN28 and its control of puberty.

Intriguingly, mammalian *MKRN3* has the opposite effect on developmental timing as its *C. elegans* ortholog *lep-2*. While *lep-2* mutant adults exhibit juvenile characteristics in the male tail tip (*Herrera et al., 2016*) and nervous system (this work), null or hypomorphic mutations in human *MKRN3* lead to Central Precocious Puberty (*Abreu et al., 2013*), in which children younger than eight (girls) or nine (boys) years of age develop secondary sexual characteristics. This may reflect a generally more complex role for a putative *MKRN3–LIN28–LET7* regulatory module in mammalian puberty compared to *C. elegans* sexual maturation. For example, overexpression of *Lin28a* delays puberty in male and female mice (*Corre et al., 2016*; *Zhu et al., 2010*), consistent with the maturation-inhibiting function of *C. elegans lin-28*. However, *Lin28b*$^{-/-}$ mice, as well as mice overexpressing *let-7*, also exhibit delayed sexual maturation, in contrast to the roles of *lin-28* and *let-7* in *C. elegans*; moreover, these effects were seen only in males (*Corre et al., 2016*). In mammals, these genes can be easily imagined to function at multiple steps in sexual development, such that apparently antagonistic activities could arise from roles for the mammalian heterochronic pathway in the production of both maturation-promoting and maturation-inhibiting signals. Clarification of these issues will require regionally and temporally controlled manipulations of mammalian gene activity. Here, we found that human *MKRN3* was not able to rescue the defects of *C. elegans lep-2* mutants. Remarkably, however, *MKRN3* overexpression caused a slight delay in the sexual maturation of two neuron types, AIM and AWA, in the *C. elegans* nervous system. This strongly suggests that the biochemical functions of LEP-2 and MKRN3 are not equivalent, and, moreover, that the mechanism by which *MKRN3* regulates pubertal timing is conserved in *C. elegans*. Several potential models are consistent with these observations (*Figure 6*). MKRN3 might inhibit LEP-2, perhaps through a dominant-negative mechanism in which MKRN3 binds to *lep-5* but is unable to recognize C. elegans LIN-28. Alternatively, MKRN3 might somehow stabilize LIN-28, rather than promoting its degradation, or, less likely, MKRN3 could function in parallel with the *lep-2*—*lin-28* module. Further, we cannot rule out the possibility that MKRN3 overexpression artifactually inhibits *lep-2* function.

Ours are not the first studies to examine roles for the heterochronic pathway in the *C. elegans* nervous system. Early in larval development, the timing of synaptic remodeling of a class of GABAergic motorneurons is controlled by the heterochronic gene *lin-14*, which specifies multiple aspects of the L1 stage (*Hallam and Jin, 1998*). Later, a hermaphrodite-specific aspect of nervous system maturation, the outgrowth of the HSN axon, is controlled by *lin-14* and *lin-28* (*Olsson-Carter and Slack, 2010*). However, in neither case have the relevant targets of heterochronic function been identified. Interestingly, the heterochronic pathway, particularly *let-7* and *lin-41*, also has a role in regulating damaged-induced regeneration of the AVM axon in adults, an ability that declines with age (*Zou et al., 2013*). Moreover, studies in *Drosophila* also support a role for the heterochronic pathway in the timing of major transitions in the nervous system (*Faunes and Larraín, 2016*). *let-7* and related miRNAs control temporal specification of neuronal fate at the larval-to-pupal transition (*Chawla et al., 2016*; *Marchetti and Tavosanis, 2017*; *Wu et al., 2012*); they also regulate remodeling of neuromuscular anatomy and function at the pupal-to-adult transition (*Caygill and Johnston, 2008*; *Sokol et al., 2008*). Recently, *Drosophila lin-28* has been found to be widely expressed in the nervous system in late third instar larvae, and loss of *lin-28* function accelerates the larval-to-pupal transition (*González-Itier et al., 2018*). Together, these findings indicate an ancient role for this regulatory module in orchestrating developmental transitions in the nervous system.

Work carried out in parallel with the studies reported here (*Pereira et al., 2019*) found that the heterochronic pathway acts cell-autonomously to control several other aspects of male-specific sexual differentiation, including a neurotransmitter fate switch and changes in synaptic connectivity. Interestingly, Pereira et al. identified *lin-29a*, a male-specific isoform of *lin-29*, as a key mediator of many of these transitions, acting to integrate temporal and sexual signals. Here, our studies consider other aspects of male-specific functional maturation and focus on upstream components of the heterochronic pathway, particularly *lep-2* and *lep-5*, best known for their roles in male tail development (*Herrera et al., 2016*; *Kiontke et al., 2019*). The strong nervous system phenotypes of these mutants indicate that this specialized branch of the heterochronic pathway is critical for the sexual maturation of the nervous system. Moreover, these studies strongly suggest that the function of *MKRN3* in regulating human puberty is related to the *LIN28–LET7* axis and predict the central involvement of a yet-unidentified lncRNA in this process.

Though all of the maturation events we have studied here are male-specific, the heterochronic pathway almost certainly has a role in hermaphrodite-specific and non-sex-specific maturation of the *C. elegans* nervous system as well. Indeed, as mentioned above, differentiation of the hermaphrodite-specific HSN neuron requires heterochronic function (*Olsson-Carter and Slack, 2010*); however, little is known about non-male-specific changes in neuronal function that accompany sexual maturation. Interestingly, two recent studies have demonstrated functional transitions in chemosensory coding and behavior that occur at the juvenile-to-adult transition in hermaphrodites (*Fujiwara et al., 2016*; *Hale et al., 2016*). In one of these cases (*Fujiwara et al., 2016*), signaling from the gonad plays an important role, raising the more general possibility that signals from the gonad could intersect with cell-autonomous timers to provide more flexible control of behavior.

Although the phenomena we have studied here derive their sex-specificity by the integration of temporal and sexual signaling at the level of *mab-3* and *lin-29a*, other interactions between sex-determination and heterochronic timing are likely. By ChIP-seq studies, *lin-28* has been identified as a direct target of *tra-1* (*Berkseth et al., 2013*); both *tra-1* and *tra-2* are potential targets of the miRNA *let-7* (targetscan.org); the abundance of TRA-1 protein has been shown to be developmentally regulated (*Weinberg et al., 2018*) (H.L. and D.S.P., in prep.); and some phenotypes of heterochronic mutants can be considered sexual transformations (*Ambros and Horvitz, 1984*). Interestingly, many aspects of the timing of mammalian puberty also appear to be sex-specific (*Abreu and Kaiser, 2016*; *Cousminer et al., 2016*). Together, these observations blur the distinctions between sexual differentiation mechanisms and developmental timing pathways, helping provide a mechanistic understanding of the observation that sexual dimorphism often arises through heterochrony (*McNamara, 2012*). Because the deep crosstalk between sexual and temporal information is unlikely to be limited to *C. elegans*, reproductive maturation disorders and sex-biased disorders of the nervous system in humans will both be informed by a deeper appreciation of these intersections.

## Materials and methods

**Key resources table**

| Reagent type (species) or resource | Designation | Source or reference | Identifiers | Additional information |
|---|---|---|---|---|
| Gene (*C. elegans*) | *him-5* | NA | WormBase ID: WBGene00001864 | |
| Gene (*C. elegans*) | *lep-2* | NA | WormBase ID: WBGene00002278 | |
| Gene (*C. elegans*) | *lep-5* | NA | WormBase ID: WBGene00010424 | |
| Gene (*C. elegans*) | *let-7* | NA | WormBase ID: WBGene00002285 | |
| Gene (*C. elegans*) | *lin-28* | NA | WormBase ID: WBGene00003014 | |

*Continued on next page*

*Continued*

| Reagent type (species) or resource | Designation | Source or reference | Identifiers | Additional information |
|---|---|---|---|---|
| Gene (C. elegans) | lin-29 | NA | WormBase ID: WBGene00003015 | |
| Gene (C. elegans) | lin-4 | NA | WormBase ID: WBGene00002993 | |
| Gene (C. elegans) | lin-41 | NA | WormBase ID: WBGene00003026 | |
| Gene (C. elegans) | mab-3 | NA | WormBase ID: WBGene00003100 | |
| Gene (C. elegans) | pha-1 | NA | WormBase ID: WBGene00004010 | |
| Gene (C. elegans) | unc-31 | NA | WormBase ID: WBGene00006767 | |
| Gene (human) | MKRN3 | NA | HGNC:7114 | |
| Strain, strain background (C. elegans) | otIs355[rab-3p(prom1):: 2xNLS::TagRFP] IV; him-5(e1490) V; nyEx62[Plep-5::GFP + pRF4] | this paper | DF305 | otIs355: OH10689; e1490: CB4088; nyEx62 see below |
| Strain, strain background (C. elegans) | lin-4(e912) II; otIs355[rab-3p(prom1):: 2xNLS::TagRFP] IV ; him-5(e1490) V; nyEx62[Plep-5::GFP + pRF4] | this paper | DF306 | e912: CB2627; otIs355: OH10689; e1490: CB4088; nyEx62 see below |
| Strain, strain background (C. elegans) | pkd-2(sy606) IV; him-5(e1490) V | Caenorhabditis Genetics Center | PT8 | RRID:WB-STRAIN:PT8 |
| Strain, strain background (C. elegans) | fsIs5[Psrj-54::YFP + cc::GFP] him-5(e1490) V | *Lee and Portman, 2007* | UR219 | |
| Strain, strain background (C. elegans) | him-5(e1490) V; kyIs53[Podr-10::odr-10::GFP] X | *Ryan et al., 2014* | UR460 | |
| Strain, strain background (C. elegans) | unc-31(e169) IV; him-5(e1490) V | PMID: 22023935 | UR624 | |
| Strain, strain background (C. elegans) | mab-3(e1240) II; him-5(e1490) V; kyIs53[ODR-10::GFP] X | this paper | UR626 | e1240: UR278; e1490: UR926; kyIs53: CX3344 |
| Strain, strain background (C. elegans) | pha-1(e2123) III; him-5(e1490) V; fsEx295[ODR-10::GFP fosmid + pBx1] | *Ryan et al., 2014* | UR773 | |
| Strain, strain background (C. elegans) | him-5(e1490) V; lin-41(bx42) X; fsEx295[ODR-10::GFP fosmid + pBx1] | this paper | UR786 | e1490: UR926; bx42: EM106; fsEx295: UR773 |
| Strain, strain background (C. elegans) | lin-41(bx42) I; fsIs5[Psrj-54::YFP + cc::GFP] him-5(e1490) V | this paper | UR869 | bx42: EM106; fsIs5: UR218; e1490: UR926 |
| Strain, strain background (C. elegans) | lin-41(ma104) I; myIs4[Ppkd-2::pkd-2:: GFP + cc::GFP] him-5 (e1490) V | this paper | UR871 | ma104: CT8; myIs4: UR1258; e1490: UR1258 |
| Strain, strain background (C. elegans) | lin-28(n719) I; fsIs5[Psrj-54::YFP + cc::GFP] him-5(e1490) V | this paper | UR872 | n719: MT1524; fsIs5: UR218; e1490: UR218 |

Continued

| Reagent type (species) or resource | Designation | Source or reference | Identifiers | Additional information |
|---|---|---|---|---|
| Strain, strain background (C. elegans) | lin-28(n719) I; myIs4[Ppkd-2::pkd-2::GFP + cc::GFP] him-5 (e1490) V | this paper | UR873 | n719: MT1524; myIs4: UR1258; e1490: UR1258 |
| Strain, strain background (C. elegans) | fsIs5[Psrj-54::YFP + cc::GFP] him-5(e1490) V; lep-5(fs8) X | this paper | UR874 | fsIs5: UR218; e1490: UR218; fs8: UR290 |
| Strain, strain background (C. elegans) | myIs4[Ppkd-2::pkd-2::GFP + cc::GFP] him-5(e1490) V; lep-5(fs8) X | this paper | UR876 | myIs4: UR1258; e1490: UR1258; fs8: UR290 |
| Strain, strain background (C. elegans) | myIs4[Ppkd-2::pkd-2::GFP + cc::GFP] him-5(e1490) V; let-7(n2853) X | this paper | UR878 | myIs4: UR1258; e1490: UR1258; n2853: MT7626 |
| Strain, strain background (C. elegans) | him-5(e1490) V | this paper | UR926 | e1490: CB4088 |
| Strain, strain background (C. elegans) | fsIs5[Psrj-54::YFP + cc::GFP] him-5(e1490) V; lep-5(fs20) X | this paper | UR1248 | fsIs5: UR218; e1490: UR218; fs20: UR1142 |
| Strain, strain background (C. elegans) | lep-2(ok900) IV; fsIs5[Psrj-54::YFP + cc::GFP] him-5(e1490) V | this paper | UR1249 | ok900: RB986; fsIs5: UR218; e1490: UR218 |
| Strain, strain background (C. elegans) | fsIs5[Psrj-54::YFP + cc::GFP] him-5(e1490) V; let-7(n2853) X | this paper | UR1250 | fsIs5: UR218; e1490: UR218; n2853: MT7626 |
| Strain, strain background (C. elegans) | pha-1(e2123) III; fsIs5[Psrj-54::YFP + cc::GFP] him-5(e1490) V; fsEx295[ODR-10::GFP fosmid + pBx1] | this paper | UR1251 | e2123: GE24; e1490: UR926; fsEx295: UR773 |
| Strain, strain background (C. elegans) | pha-1(e2123) III; lep-2(ok900) IV; him-5(e1490) V; fsEx295[ODR-10::GFP fosmid + pBx1] | this paper | UR1252 | e2123: GE24; ok900: RB986; e1490: UR926; fsEx295: UR773 |
| Strain, strain background (C. elegans) | lin-28(n719) I; pha-1(e2123) III; him-5(e1490) V; fsEx295[ODR-10::GFP fosmid + pBx1] | this paper | UR1253 | n719: MT1524; e2123: GE24; e1490: UR926; fsEx295: UR773 |
| Strain, strain background (C. elegans) | him-5(e1490) V; let-7(n2853) X; fsEx295[ODR-10::GFP fosmid + pBx1] | this paper | UR1254 | e1490: UR926; n2853: MT7626; fsEx295: UR773 |
| Strain, strain background (C. elegans) | pha-1(e2123) III; him-5(e1490) V; let-41(ma104) X; fsEx295[ODR-10::GFP fosmid + pBx1] | this paper | UR1255 | e2123: GE24; e1490: UR926; ma104: CT8; fsEx294: UR773 |
| Strain, strain background (C. elegans) | lin-29a(xe38) II; him-5(e1490) V; kyIs53[Podr-10::odr-10::GFP] X | this paper | UR1256 | xe38: HW1693; e1490: UR926; kyIs53: CX3344 |
| Strain, strain background (C. elegans) | lin-29a(xe40) II; him-5(e1490) V; kyIs53[Podr-10::odr-10::GFP] X | this paper | UR1257 | xe40: HW1695; e1490: UR926; kyIs53: CX3344 |

Continued

| Reagent type (species) or resource | Designation | Source or reference | Identifiers | Additional information |
|---|---|---|---|---|
| Strain, strain background (*C. elegans*) | *myIs4[Ppkd-2::pkd-2::GFP + cc::GFP] him-5 (e1490) V* | this paper | UR1258 | *myIs4*: PT621; *e1490*: PT621 |
| Strain, strain background (*C. elegans*) | *myIs4[Ppkd-2::pkd-2::GFP + cc::GFP] him-5(e1490) V; lep-5(ny28) X* | this paper | UR1259 | *myIs4*: PT621; *e1490*: PT621; *ny28*: UR1022 |
| Strain, strain background (*C. elegans*) | *lep-2(ok900) IV; myIs4[Ppkd-2::pkd-2::GFP + cc::GFP] him-5 (e1490) V* | this paper | UR1260 | *ok900*: RB986; *myIs4*: UR1258; *e1490*: UR1258 |
| Strain, strain background (*C. elegans*) | *lin-41(bx42) I; myIs4[Ppkd-2::pkd-2::GFP + cc::GFP] him-5 (e1490) V* | this paper | UR1261 | *bx42*: EM106; *myIs4*: UR1258; *e1490*: UR1258 |
| Strain, strain background (*C. elegans*) | *him-5(e1490) V; lep-5(ny28) X; fsEx527[1.5 kb MAB-3::GFP (pDZ162) + cc::GFP]* | this paper | UR1262 | *e1490*: UR926; *ny28*: UR1022; *fsEx527*: UR1131 |
| strain, strain background (*C. elegans*) | *lep-2(ok900) IV; him-5(e1490) V; fsEx527[1.5 kb MAB-3::GFP (pDZ162) + cc::GFP]* | this paper | UR1263 | *ok900*: RB986; *e1490*: UR926; *fsEx527*: UR1131 |
| Strain, strain background (*C. elegans*) | *lin-28(n719) I; him-5(e1490) V; fsEx527[1.5 kb MAB-3::GFP (pDZ162) + cc::GFP]* | this paper | UR1264 | *n719*: MT1524; *e1490*: UR926; *fsEx527*: UR1131 |
| Strain, strain background (*C. elegans*) | *him-5(e1490) V; ksIs2[Pdaf-7::GFP + rol-6(su1006)]* | this paper | UR1265 | *e1490*: UR926; *ksIs2*: FK181 |
| Strain, strain background (*C. elegans*) | *him-5(e1490) V; lep-5(ny28) X; ksIs2[Pdaf-7::GFP + rol-6(su1006)]* | this paper | UR1266 | *e1490*: UR926; *ny28*: UR1022; *ksIs2*: FK181 |
| Strain, strain background (*C. elegans*) | *lep-2(ok900) IV; him-5(e1490) V; ksIs2[Pdaf-7::GFP + rol-6(su1006)]* | this paper | UR1267 | *ok900*: RB986; *e1490*: UR926; *ksIs2*: FK181 |
| Strain, strain background (*C. elegans*) | *fsIs5[Psrj-54::YFP + cc::GFP] him-5(e1490) V; fsEx559[Prab-3::MKRN3::SL2::mCherry::unc-54 3' UTR + Pmyo-3::mCherry Line 1]* | this paper | UR1268 | *fsIs5*: UR218; *e1490*: UR926; *fsEx559*: see below |
| Strain, strain background (*C. elegans*) | *fsIs5[Psrj-54::YFP + cc::GFP] him-5(e1490) V; fsEx560[Prab-3::MKRN3::SL2::mCherry::unc-54 3' UTR + Pmyo-3::mCherry Line 2]* | this paper | UR1269 | *fsIs5*: UR218; *e1490*: UR926; *fsEx560*: see below |
| Strain, strain background (*C. elegans*) | *lep-2(ok900) IV; fsIs5[Psrj-54::YFP + cc::GFP] him-5(e1490) V; fsEx561[Peat-4prom11::lep-2::Sl2::mCherry::unc-54 3' UTR + Pmyo-3::mCherry Line 1]* | this paper | UR1270 | *ok900*: RB986; *fsIs5*: UR218; *e1490*: UR218; *fsEx561*: see below |

*Continued*

| Reagent type (species) or resource | Designation | Source or reference | Identifiers | Additional information |
|---|---|---|---|---|
| Strain, strain background (*C. elegans*) | *lep-2(ok900) IV; fsIs5[Psrj-54::YFP + cc::GFP] him-5(e1490) V; fsEx562[Peat-4prom11::lep-2::Sl2::mCherry::unc-54 3' UTR + Pmyo-3::mCherry Line 2]* | this paper | UR1271 | *ok900*: RB986; *fsIs5*: UR218; *e1490*: UR218; *fsEx562*: see below |
| Strain, strain background (*C. elegans*) | *lep-2(ok900) IV; fsIs5[Psrj-54::YFP + cc::GFP] him-5(e1490) V; fsEx563[Peat-4prom11::lep-2::Sl2::mCherry::unc-54 3' UTR + Pmyo-3::mCherry Line 3]* | this paper | UR1272 | *ok900*: RB986; *fsIs5*: UR218; *e1490*: UR218; *fsEx563*: see below |
| Strain, strain background (*C. elegans*) | *lep-2(ok900) IV; fsIs5[Psrj-54::YFP + cc::GFP] him-5(e1490) V; fsEx564[Prab-3::MKRN3::SL2::mCherry::unc-54 3' UTR + Pmyo-3::mCherry Line 1]* | this paper | UR1273 | *ok900*: RB986; *fsIs5*: UR218; *e1490*: UR218; *fsEx564*: see below |
| Strain, strain background (*C. elegans*) | *lep-2(ok900) IV; fsIs5[Psrj-54::YFP + cc::GFP] him-5(e1490) V; fsEx565[Prab-3::MKRN3::SL2::mCherry::unc-54 3' UTR + Pmyo-3::mCherry Line 2]* | this paper | UR1274 | *ok900*: RB986; *fsIs5*: UR218; *e1490*: UR218; *fsEx565*: see below |
| Strain, strain background (*C. elegans*) | *lin-28(n719) I; fsIs5[Psrj-54::YFP + cc::GFP] him-5(e1490) V; fsEx566[Peat-4prom11::lin-28::Sl2::mCherry::unc-54 3' UTR + Pmyo-3::mCherry Line 1]* | this paper | UR1275 | *n719*: MT1524; *fsIs5*: UR218; *e1490*: UR218; *fsEx566*: see below |
| Strain, strain background (*C. elegans*) | *lin-28(n719) I; fsIs5[Psrj-54::YFP + cc::GFP] him-5(e1490) V; fsEx567[Peat-4prom11::lin-28::Sl2::mCherry::unc-54 3' UTR + Pmyo-3::mCherry Line 2]* | this paper | UR1276 | *n719*: MT1524; *fsIs5*: UR218; *e1490*: UR218; *fsEx567*: see below |
| Strain, strain background (*C. elegans*) | *pha-1(e2123) III; lep-2(ok900) IV; him-5(e1490) V; fsEx295[ODR-10::GFP fosmid + pBx1]; fsEx568[Pgpa-4del6::lep-2::Sl2::mCherry::unc-54 3' UTR + Pelt-2::GFP Line 1]* | this paper | UR1277 | *pha-1*: GE24; *ok900*: RB986; *e1490*: UR926; *fsEx295*: UR773; *fsEx568*: see below |
| Strain, strain background (*C. elegans*) | *pha-1(e2123) III; lep-2(ok900) IV; him-5(e1490) V; fsEx295[ODR-10::GFP fosmid + pBx1]; fsEx569[Pgpa-4del6::lep-2::Sl2::mCherry::unc-54 3' UTR + Pelt-2::GFP Line 2]* | this paper | UR1278 | *e2123*: GE24; *ok900*: RB986; *e1490*: UR926; *fsEx295*: UR773; *fsEx569*: see below |
| Strain, strain background (*C. elegans*) | *lin-28(n719) I; lep-2(ok900) IV; fsIs5[Psrj-54::YFP + cc::GFP] him-5(e1490) V* | this paper | UR1279 | *n719*: MT1524; *ok900*: RB986; *fsIs5*: UR218; *e1490*: UR218 |
| Strain, strain background (*C. elegans*) | *lin-28(n719) I; fsIs5[Psrj-54::YFP + cc::GFP] him-5(e1490) V; lep-5(ny28) X* | this paper | UR1280 | *n719*: MT1524; *fsIs5*: UR218; *e1490*: UR218; *ny28*: UR1022 |
| Strain, strain background (*C. elegans*) | *lin-28(n719) I; fsIs5[Psrj-54::YFP + cc::GFP] him-5(e1490) V; let-7(n2853) X* | this paper | UR1281 | *n719*: MT1524; *fsIs5*: UR218; *e1490*: UR218; *n2853*: MT7626 |

*Continued on next page*

*Continued*

| Reagent type (species) or resource | Designation | Source or reference | Identifiers | Additional information |
|---|---|---|---|---|
| Recombinant DNA reagent | MKRN3 (NM_005664) Human Untagged Clone | Origene | Cat# SC319872 | |
| Recombinant DNA reagent | pDONR221 | Invitrogen | | https://www.addgene .org/vector-database/2394/ |
| Recombinant DNA reagent | *Peat-4prom11::lep-2::SL2::mCherry::unc-54 3′UTR* | this paper | | Construction described below |
| Recombinant DNA reagent | *Peat-4prom11::lin-28::SL2::mCherry::unc-54 3′UTR* | this paper | | Construction described below |
| Recombinant DNA reagent | *Pgpa-4del6::lep-2::SL2::mCherry::unc-54 3′UTR* | this paper | | Construction described below |
| Recombinant DNA reagent | *Prab-3::MKRN3::SL2::mCherry::unc-54 3′UTR* | this paper | | Construction described below |
| Software, algorithm | Prism 8 | | GraphPad Software | RRID:SCR_002798 |
| Software, algorithm | ApE, A Plasmid Editor | | M Wayne Davis | RRID:SCR_014266 http://jorgensen.biology .utah.edu/wayned/ape/ |

## *C. elegans* culture and staging

In general, *C. elegans* were cultured using standard methods (*Stiernagle, 2006*). All strains used here carried the mutation *him-5(e1490)* to increase the numbers of males, such that we refer to *him-5(e1490)* as the wild-type. All strains used in this work are listed in the Key resources table..

To isolate populations at specific larval stages, adult hermaphrodites were allowed to lay eggs for two hours, roughly synchronizing their progeny. Because male tail morphogenesis cannot be used to reliably stage some heterochronic mutant males under low magnification, sibling hermaphrodites were used as proxies to roughly stage animals. For all individuals scored, larval stage was definitively determined by the progression of gonadal development using DIC microscopy (Zeiss Axioplan 2, 63x Plan-Apo objective). Males in which the linker cell had migrated dorsally but not yet ventrally were staged as L3 larvae. Males in which the linker cell had migrated ventrally and extended posteriorly were staged as L4 larvae. One-day old adults were obtained by picking L3-L4 larvae and allowing them to mature overnight. Because *lin-28* hermaphrodites have severe egg-laying defects, we passed all strains containing *lin-28* mutations through the dauer stage to suppresses this phenotype (*Liu and Ambros, 1991*), allowing us to use timed egg-laying for rough synchronization.

## Imaging and quantitation of *fsEx295*, *fsIs5*, *fsEx527*, *ksIs2*, and *myIs4*

Animals were mounted on 4% agarose pads in M9 buffer supplemented with levamisole. GFP and DIC brightfield images were obtained using a 63x PlanApo objective on a Zeiss Axioplan 2. GFP intensity was qualitatively determined to be bright (3), moderate (2), faint (1), or absent (0). For rescue and overexpression experiments, the experimenter was blind to the presence or absence of the expression array when scoring reporter fluorescence. Animals not bearing the array were used as non-transgenic sibling controls. Representative images were taken using the same illumination intensity and exposure time. Images of *ksIs2* were captured using an Apotome. Fluorescence intensity data were compared using Mann-Whitney tests (for pairwise analyses) or Kruskal-Wallis tests with Dunn's correction (for multiple comparisons). For these and all other statistical analyses reported in this work, statistical significance is indicated in the figures as follows: *$0.01 < p < 0.05$; **$0.001 < p < 0.01$; ***$p < 0.001$.

## Imaging of strains containing *maIs108* and *nyEx53*

Animals were mounted as described above and images were obtained using a Leica SP5 Confocal Microscope with a HyD hybrid detector and 63x objective. Z-series were obtained starting at the

most superficial neuron and ending at the deepest neuron as determined by expression of the pan-neural nuclear marker *otIs355[Prab-3::tagRFP::2xNLS]*. Images were taken at 0.5 μm steps, 200 Hz, and 2x zoom. FIJI (*Schindelin et al., 2012*) was used to create max-intensity z-projections of the entire z-stack. Images were pseudocolored and merged using Adobe Photoshop. Quantitation and statistical analysis was carried out as described in the preceding paragraph.

## Imaging of strains containing *nyEx62*

Animals were mounted as above but were anesthetized with 20 mM $NaN_3$. Images were obtained using a Zeiss AxioImager equipped with an Apotome using the 40x objective. Ten to fifteen slices, taken with a spacing of 0.5 μm, were merged into a Z projection using Image J.

## Transgenic strains

cDNAs were amplified from RNA extracted from mixed-stage *him-5* cultures. *lep-2* and *lin-28* cDNAs were amplified using primers shown in Table S1. Human *MKRN3* cDNA was amplified from the plasmid SC319872 (Origene) using primers shown in Table S1. The resulting cDNAs were cloned into pDONR221 and recombined via Multisite Gateway Cloning (Invitrogen) to make following expression clones:

> Peat-4prom11::lep-2::SL2::mCherry::unc-54 3′UTR
> Peat-4prom11::lin-28::SL2::mCherry::unc-54 3′UTR
> Pgpa-4del6::lep-2::SL2::mCherry::unc-54 3′UTR
> Prab-3::MKRN3::SL2::mCherry::unc-54 3′UTR

Transgenes were injected into young adult hermaphrodites using standard methods (*Evans, 2006*) to generate *fsEx* transgenic lines.

The *Plep-5::GFP* reporter was constructed by overlap extension PCR (*Nelson and Fitch, 2011*) using primers KKOlp-5_1 and plep-5_GFP_B to amplify *C. elegans* genomic DNA (for fragment A) and primers pPD95.75.C and pPD95.75D to amplify from plasmid pPD95.75 (for fragment B). Final extension used primers KKlep5_expr-9 and pPD95.75.D*. Transgenes were injected with standard methods using the co-injection marker pRF4[*rol-6(d)*].

## Behavioral assays

### Food-Leaving assay

Males were isolated to a newly seeded plate approximately 16 hr prior to the start of the assay, at the L4 stage (to assay adults) or L1 (to assay L3s). Single males were placed on a 7 μL spot of *E. coli* OP50 in the center of a 10 cm plate containing 10 mL of standard agar lacking cholesterol based on the method of *Lipton et al. (2004)*. At designated intervals, tracks were examined to determine the farthest distance traveled by the animal: zero (never left food),<1 cm (a 'minor excursion' off the food),>1 cm but <3.5 cm (a 'major excursion' off the food), or >3.5 cm (left the plate). Animals that died on the plate were censored. The fraction in each category for each genotype or age was compared with controls using a chi-square test (Prism 7.0).

### Ascaroside attraction

Pheromone attraction assays were performed and data were analyzed as previously described (*Fagan et al., 2018*).

### Response and location of vulva

Adult males were placed on a mating assay plate with 15 to 20 *unc-31; him-5* hermaphrodites and allowed four minutes to display mating behavior as previously described (*Barr and Sternberg, 1999*; *Liu and Sternberg, 1995*), Males were assayed in pairs and watched in real time to determine whether they could (1) respond to hermaphrodite contact and (2) locate the vulva. Data is reported as follows: Response = # of males responding/total # of males assayed. Location = # of males that locate the vulva/total # of males that responded. For each step, successful individuals were scored as one and unsuccessful individuals as 0. Kruskal-Wallis tests (with Dunn's correction) were used to analyze differences between groups.

## Acknowledgements

We are grateful to Jintao Luo for help with strain construction, Victoria DiMarco and Adam Mason for preliminary studies on *srj-54*, Laura Pereira and Oliver Hobert for sharing unpublished results, and Florian Aeschimann and Helge Großhans for providing *lin-29a* alleles. Some strains used in this work were provided by the *Caenorhabditis* Genetics Center (University of Minnesota), which is funded by the NIH Office of Research Infrastructure Programs (P40 OD010440). This research was supported by NIH R01 GM108885, R01 GM130136, and NSF IOS 1353075 (to DSP) and by NIH R01 GM100140, NSF DEB 0922012, and research funds from NYU Shanghai (to DHAF). EV was supported by the University of Rochester Medical Scientist Training Program, NIH T32 GM007356.

## Additional information

### Funding

| Funder | Grant reference number | Author |
| --- | --- | --- |
| National Institute of General Medical Sciences | R01 GM108885 | Douglas Portman |
| National Institute of General Medical Sciences | R01 GM130136 | Douglas Portman |
| National Science Foundation | IOS 1353075 | Douglas Portman |
| National Institute of General Medical Sciences | T32 GM007356 | Edward Vuong |
| National Institute of General Medical Sciences | R01 GM100140 | David Fitch |
| National Science Foundation | DEB 0922012 | David Fitch |
| New York University | | David Fitch |

The funders had no role in study design, data collection and interpretation, or the decision to submit the work for publication.

### Author contributions

Hannah Lawson, Conceptualization, Resources, Formal analysis, Investigation, Writing—original draft; Edward Vuong, Conceptualization, Investigation; Renee M Miller, Conceptualization, Formal analysis, Investigation; Karin Kiontke, Conceptualization, Resources, Investigation; David HA Fitch, Conceptualization, Resources, Funding acquisition; Douglas S Portman, Conceptualization, Formal analysis, Supervision, Funding acquisition, Writing—review and editing

### Author ORCIDs

Douglas S Portman (iD) https://orcid.org/0000-0003-2686-8839

### Decision letter and Author response

Decision letter https://doi.org/10.7554/eLife.43660.021
Author response https://doi.org/10.7554/eLife.43660.022

## Additional files

### Supplementary files

• Transparent reporting form
DOI: https://doi.org/10.7554/eLife.43660.019

### Data availability

Data is available in supporting files.

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
