## [Decision Letter]

Thank you for submitting your article "The Makorin *lep-2* and the lncRNA *lep-5* regulate *lin-28* to schedule sexual maturation of the *C. elegans* nervous system" for consideration by *eLife*. Your article has been reviewed by two peer reviewers, and the evaluation has been overseen by a Reviewing Editor and Catherine Dulac as the Senior Editor. The following individual involved in review of your submission has agreed to reveal his identity: L René García (Reviewer #2).

The reviewers have discussed the reviews with one another and the Reviewing Editor has drafted this decision to help you prepare a revised submission.

Summary:

The manuscript, "The Makorin *lep-2* and the IncRNA *lep-5* regulate *lin-28* to schedule sexual maturation of the *C. elegans* nervous system", demonstrates that the *lin-28/let-7/lin-41* heterochronic gene pathway is used to direct the developmentally programmed sexual dimorphism of *C. elegans* male neurons. In previous work, the authors have beautifully worked out the genetics that describe the molecular regulation of sexual dimorphism in neurons common to male and hermaphrodite *C. elegans*. In this manuscript, the research builds on this foundation to identify the pathway that determines when sexual dimorphism is expressed. The work uses genetic analysis to place *lep-2*, a homolog of mammalian Makorin, and *lep-5*, a long noncoding RNA, upstream of *lin-28*. Assayed neurons in *lep-2* and *lep-5* mutants show delayed transition from larval to adult gene expression changes. Consistent with the retardation in developmental sexual dimorphism, the adult *lep* mutants show abnormal copulatory drive characteristics. The researchers also show that LEP-2 and LIN-28 regulate the timing of neural sexual remodeling cell-autonomously, which is an important observation. Finally, the researchers make an argument that the mammalian Makorin's function, as a regulator that delays reproductive maturity, can be capitulated experimentally in *C. elegans*.

General assessment:

Both reviewers find the work interesting and important, especially praising the finding that LEP-2 and LIN-28 act cell-autonomously. However, both of them point out the need for clarification in several points.

Essential revisions:

1) Both reviewers raise concern about *MKRN3*, because in the *MKRN3* overexpression experiments the authors looked at only changes in *srj-54* expression in AIM and the effect seen here was rather modest. The reviewers request looking at other maturation marker genes in other neurons. These experiments need to be performed upon revision. Specifically,

- Regarding the experiment showing that expression of human *MKRN3* in worms delays sexual maturation, it would be nice to see that this is a general effect that occurs with more than one marker in more than one neuron (the authors only show delay in *srj-54* expression in AIM).

- In the last paragraph of the subsection “Human *MKRN3* can inhibit sexual differentiation of the *C. elegans* nervous system”, the authors state that transgenic expression of *MKRN3*, has a statistical effect of delaying adult male gene expression in AIM during L4 to adult stage. For AIM, the effect is modest, which is ok, but I do not feel by itself is strong enough to support the claim that Makorin's function in mammals can be recapitulated in the *C. elegans* model. If one can show data for modest (statistical) retardation in developmental gene expression for the other neurons studied in the paper: AWA(*odr-10*), ADF(*mab-3*), ASJ (*daf-7*), CEM (*pkd-2*), then I would feel more confident that *MKRN3* generally affects the worm's heterochronic pathway in many neurons. One might need to use stronger promoters than the rab-3 promoter for each neuron tested, but this would provide more robust evidence for placing Makorin in the model presented in Figure 6.

2) One of the reviewers points out the need for quantification of the results in Figure 4—figure supplement 1. Indeed, while quantitative data are shown for Figure 4—figure supplement 1D, it is not the case for Figure 4—figure supplement 1A and B. Quantification of expression levels and qualitative description of the expression pattern need to be added.

- Supplementary Figure 2A needs quantification or more details about the phenotype: is it faint in 100% males (as the image shows) or what kind of *srj-54* phenotype do *mab-3* mutants display?

---

## [Author Response]

Essential revisions:1) Both reviewers raise concern about MKRN3, because in the MKRN3 overexpression experiments the authors looked at only changes in srj-54 expression in AIM and the effect seen here was rather modest. The reviewers request looking at other maturation marker genes in other neurons. These experiments need to be performed upon revision. Specifically,- Regarding the experiment showing that expression of human MKRN3 in worms delays sexual maturation, it would be nice to see that this is a general effect that occurs with more than one marker in more than one neuron (the authors only show delay in srj-54 expression in AIM).- In the last paragraph of the subsection “Human MKRN3 can inhibit sexual differentiation of the C. elegans nervous system”, the authors state that transgenic expression of MKRN3, has a statistical effect of delaying adult male gene expression in AIM during L4 to adult stage. For AIM, the effect is modest, which is ok, but I do not feel by itself is strong enough to support the claim that Makorin's function in mammals can be recapitulated in the C. elegans model. If one can show data for modest (statistical) retardation in developmental gene expression for the other neurons studied in the paper: AWA(odr-10), ADF(mab-3), ASJ (daf-7), CEM (pkd-2), then I would feel more confident that MKRN3 generally affects the worm's heterochronic pathway in many neurons. One might need to use stronger promoters than the rab-3 promoter for each neuron tested, but this would provide more robust evidence for placing Makorin in the model presented in Figure 6.

We agree that our analysis of the *MKRN3* overexpression phenotype was somewhat preliminary. In response to these concerns, the revised paper now includes data for two additional markers, *odr-10* (AWA) and *pkd-2* (CEM). We find that *MKRN3* overexpression causes a significant delay in the temporally controlled downregulation of *odr-10* expression, strengthening our initial conclusion that *MKRN3* can inhibit *C. elegans* nervous system maturation. In contrast, we observed no detectable effect of *MKRN3* on *pkd-2* expression, raising the possibility that it may not regulate CEM maturation. This is consistent with our earlier observation that CEM maturation seems to rely less on *lep-2* and *lep-5* than does AWA maturation (Figure 2) and that the CEMs therefore likely employ other mechanisms to time their differentiation.

For technical reasons, we were unable to score expression of *mab-3* or *daf-7* in our *MKRN3* overexpression strains. The transmission of our *mab-3::GFP* transgene was too low for us to be able to reliably assess expression in the simultaneous presence of the *MKRN3* arrays. Furthermore, we were unable to obtain stable lines carrying both the *daf-7* reporter transgene (*ksIs2*) and a *MKRN3* overexpression array. While the reasons for this are unknown, we feel that our scoring of *odr-10* and *pkd-2* substantially strengthens our analysis of the *MKRN3* phenotype. Our paper now demonstrates that human *MKRN3* can inhibit two distinct aspects of the maturation of the *C. elegans* nervous system (*srj-54*/AIM and *odr-10*/AWA), but that other aspects (*pkd-2*/CEM) may not be influenced by this mechanism.

2) One of the reviewers points out the need for quantification of the results in Figure 4—figure supplement 1. Indeed, while quantitative data are shown for Figure 4—figure supplement 1D, it is not the case for Figure 4—figure supplement 1A and B. Quantification of expression levels and qualitative description of the expression pattern need to be added.- Supplementary Figure 2A needs quantification or more details about the phenotype: is it faint in 100% males (as the image shows) or what kind of srj-54 phenotype do mab-3 mutants display?

We agree with these points. The revised manuscript now reports careful quantitative scoring of both *srj-54* and *odr-10* expression in *mab-3* mutants. In agreement with our previous qualitative findings, this reveals that there are significant defects in the adult expression of both markers.